# Astrocytic insulin receptor controls circadian behavior via dopamine signaling in a sexually dimorphic manner

Antía González-Vila[1,2,8], María Luengo-Mateos[1,8], María Silveira-Loureiro ®[1,2], Pablo Garrido-Gil ®[3,4], Nataliia Ohinska[1,5], Marco González-Domínguez ®[1], Jose Luis Labandeira-García[3,4], Cristina García-Cáceres[6,7], Miguel López ®[2] ✉ & Olga Barca-Mayo ®[1] ✉

Mammalian circadian clocks respond to feeding and light cues, adjusting internal rhythms with day/night cycles. Astrocytes serve as circadian time-keepers, driving daily physiological rhythms; however, it's unknown how they ensure precise cycle-to-cycle rhythmicity. This is critical for understanding why mistimed or erratic feeding, as in shift work, disrupts circadian physiology- a condition linked to type 2 diabetes and obesity. Here, we show that astrocytic insulin signaling sets the free-running period of locomotor activity in female mice and food entrainment in male mice. Additionally, ablating the insulin receptor in hypothalamic astrocytes alters cyclic energy homeostasis differently in male and female mice. Remarkably, the mutants exhibit altered dopamine metabolism, and the pharmacological modulation of dopaminergic signaling partially restores distinct circadian traits in both male and female mutant mice. Our findings highlight the role of astrocytic insulin-dopaminergic signaling in conveying time-of-feeding or lighting cues to the astrocyte clock, thus governing circadian behavior in a sex-specific manner.

The circadian timekeeping system is a network of brain clocks and peripheral oscillators that enable mammals to adapt to periodic daily events, such as light/dark (LD) cycles and food availability[1,2]. In mammals, circadian rhythmicity is modulated by the hypothalamic supra-chiasmatic nucleus (SCN), which coordinates the clocks throughout the body with the daily LD cycle[1,2]. In peripheral tissues, circadian clocks are entrained by daily cycles of food availability, which can uncouple circadian oscillators in peripheral tissues from the SCN[3–5]. Time-restricted feeding during the inactive period is associated with a

shift in peripheral oscillators, such as the liver, and with the rise of a daily bout, namely food anticipation activity (FAA), that precedes meal time[3–5]. FAA, exhibits the formal properties of a circadian clock, can be uncoupled from the SCN, which remains synchronized to the light[3–5] and is preserved in the absence of a functional SCN[6–8]. Therefore, the entrainment to food availability has been proposed to be driven by a food-entrainable oscillator (FEO) of a still unknown location[8]. The mechanism underlying entrainment to feeding time is critical for understanding why mistimed feeding and light exposure, as occurs

[1]Circadian and Glial Biology Lab, Physiology Department, Molecular Medicine and Chronic Diseases Research Centre (CiMUS), University of Santiago de Compostela, Santiago de Compostela, Spain. [2]NeurObesity Lab, Physiology Department, Molecular Medicine and Chronic Diseases Research Centre (CiMUS), University of Santiago de Compostela, Santiago de Compostela, Spain. [3]Laboratory of Cellular and Molecular Neurobiology of Parkinson's Disease, Department of Morphological Science, Molecular Medicine and Chronic Diseases Research Centre (CiMUS), University of Santiago de Compostela, Santiago de Compostela, Spain. [4]Networking Research Center on Neurodegenerative Diseases (CIBERNED), Madrid, Spain. [5]Horbachevsky Ternopil National Medical University, Ternopil, Ukraine. [6]Institute for Diabetes and Obesity, Helmholtz Diabetes Center, Helmholtz Munich & German Center for Diabetes Research (DZD), 85764 Neuherberg, Germany. [7]Medizinische Klinik und Poliklinik IV, Klinikum der Universität, Ludwig-Maximilians-Universität München, 80336 Munich, Germany. [8]These authors contributed equally: Antía González-Vila, María Luengo-Mateos. ✉e-mail: m.lopez@usc.es; olga.barca.mayo@usc.es

during shift work, disrupts circadian physiology[9,10]. This condition is associated with an increased incidence of chronic diseases, such as type 2 diabetes and metabolic syndrome[11,12]. However, the cellular and molecular mechanisms of normal and pathological entrainment via feeding signals are not yet fully understood.

At the molecular level, circadian rhythms are generated by an autoregulatory transcriptional/translational feedback loop (TTFL) of clock genes/proteins[13]. The E-box-specific transcription factors Basic Helix-Loop-Helix ARNT Like 1 (BMAL1) and Circadian Locomotor Output Cycles Protein Kaput (CLOCK) are the positive limb of the TTFL, which bind as heterodimers to E-boxes present in the promoters of target genes, such as *Period* (*Per1, 2, 3*) and *Cryptochrome* (*Cry1, 2*), to activate their transcription[14–16]. The negative loop comprises the PER/CRY complex that, upon accumulation, leads to the degradation of BMAL1/CLOCK dimers, thus inhibiting their transcription[17]. This feedback loop is responsible for the oscillation of about 10–20% of all genes expressed[18,19]. However, 20% of oscillating proteins do not show signs of rhythmicity at the mRNA level. This is relevant, as it suggests that the transcriptional feedback loop is not the only mechanism for generating rhythmicity in living organisms[20]. The precision of this molecular clock is assured by many post-transcriptional modifications, translational regulation, alterations in protein stability, and subcellular localization changes among the protein constituents of the circadian system[21]. For example, insulin (INS) signaling, which regulates anabolic processes triggered during feeding, reset circadian clocks in vivo and in vitro by induction of PERIOD proteins[22].

Recent evidence has shown that populations of coupled SCN neurons and astrocytes function as master pacemakers responsible for coordinating circadian oscillators in the brain and peripheral tissues to the daily LD cycle[23–29]. However, it is still being determined how the astrocyte clock is entrained to lead to the cycle-to-cycle precision of circadian rhythmicity in the SCN and/or extra SCN brain clocks. This knowledge would be crucial for adjusting the astrocyte clock and mitigating the alteration of the range of physiological processes affected by circadian disruption, such as such as those in shift workers[9–12]. Astrocytes are pivotal for regulating energy homeostasis, crucial in nutrient sensing[2], and express receptors for hormones such as INS[30,31]. However, whether astrocytic INS signaling is involved in light and food entrainment is unknown. Here, we show that astrocytic INS signaling plays a crucial role in the light and food entrainment of circadian behavior through a dopamine-mediated, sex-dependent mechanism.

## Results

### Insulin (INS) synchronizes the molecular clock in primary astrocytes

Astrocytes are competent circadian oscillators[32] whose rhythms can be synchronized in vitro by glucocorticoids[25,33]. We analyzed gene expression data of dexamethasone (DEX)-treated mouse cortical astrocytes to assess the intrinsic astrocyte clock's impact on global gene expression[34]. We identified circadian transcripts exhibiting near 24-h rhythms with a 5% false-discovery rate (FDR) for detection. In DEX-stimulated astrocytes, 5310 probes were classified as circadian, targeting 2881 unique genes and 2429 transcripts of unknown function (Supplementary Fig. 1a and Supplementary Data 1). The percentage of rhythmic transcripts, at 11.72% of the total detected probe sets, aligns with previous reports for various tissues in vivo[35,36]. Gene ontology (GO) analyses revealed biological processes linked to astrocyte circadian transcripts, encompassing localization, development, metabolism, and rhythmic processes (Supplementary Fig. 1b and Supplementary Data 2).

Astrocyte circadian transcripts exhibited peaks at 9 h (725 transcripts), 15 h (754 transcripts), and 20 h (1647 transcripts) post DEX treatment, suggesting daily "rush hours" of transcription

(Supplementary Fig. 1c). Cluster analysis subdivided these circadian transcripts into four peaking clusters (Supplementary Fig. 1a). Notably, GO analyses revealed significant segregation of transcripts' phases concerning specific KEGG pathways (Supplementary Fig. 1d). The first cluster, among others, encompassed phototransduction-associated KEGG pathways. The second cluster was enriched in cell cycle and proliferation pathways, while the third cluster was linked to lipid metabolism and glucocorticoid-related signaling. Significantly, the fourth cluster included KEGG pathways such as INS signaling and dopaminergic synapse (Supplementary Fig. 1d and Supplementary Data 3 and 4). This implies circadian INS receptor (IR) sensitivity regulation in astrocytes, aligned with reported brain glucose utilization patterns-higher during the dark phase[37,38]. Notably, the SCN contrasts with elevated light-phase glucose utilization[37,39,40]. Recent findings suggest the SCN manages day-night glycemia variations via vasopressin, impacting glucose transporter 1 (GLUT1) expression in the hypothalamic arcuate nucleus (ARC). As GLUT1 is expressed by astrocytes[41] this may affect hypothalamic INS sensitivity and glucose use. Investigating further, we qRT-PCR analyzed hypothalamic IR and glucose transporter expression across circadian times. All transporters investigated (*Glut1*, *Glut2*, and *Glut4*) exhibited rhythmic oscillations, peaking around ZT12-ZT18, yet IR showed no oscillation (Supplementary Fig. 1e). Additionally, hypothalamic INS signaling was elevated at ZT0, as evidenced by phosphorylated AKT (pAKT), aligning with circadian regulation of brain IR sensitivity (Supplementary Fig. 1f). These findings support the diurnal control of hypothalamic IR sensitivity and suggest that astrocytes may play a key role in this regulatory mechanism.

Notably, INS resets circadian clock oscillations in vivo and different cell types in vitro[22,42–44]. Yet, whether it synchronizes the astrocyte's clock remains unknown. To investigate this, we treated primary hypothalamic astrocytes with a 1-h INS or DEX pulse in low glucose medium, using DEX as a positive control[25,33]. Our results showed both DEX and INS synchronized *Per2, Cry1, Bmal1* and *Dbp* (Fig. 1a). However, DEX stimulation in astrocytes delayed the phase of *Per2, Cry1*, and *Dbp* by around 4 h compared to INS treatment (Fig. 1b).

Daily glucose level oscillations have been observed to influence clock gene expression and transcriptional outputs in the hypothalamus[45]. Additionally, glucose uptake is a result of INS signaling[46], and cellular circadian rhythms respond to glucose availability[47,48]. To mimic in vivo feeding-related changes, we combined glucose (high glucose medium) with INS administration (Fig. 1a). Surprisingly, glucose presence reduced *Per2* amplitude and advanced *Per2, Bmal1*, and *Dbp* phases (Fig. 1a, b). Previous studies showed that DEX induces *Per2* expression[49,50], while INS induces rapidly translates existing *Per2* mRNA[22]. Additionally, INS[51] and glucose[52] influence CRY1 expression and/or degradation. Thus, we assessed PER2 and CRY1 expression in primary hypothalamic astrocytes synchronized with INS under low and high glucose. Our findings showed rhythmic oscillations in both proteins (Fig. 1c), consistent with *Per2* and *Cry1* mRNA levels (Fig. 1a). To differentiate between transcriptional and translational control of the astrocyte's clock, we assessed *Per2, Cry1, Bmal1*, and *Dbp* mRNA levels in primary hypothalamic astrocytes under low glucose conditions at shorter time points after DEX or INS treatment (1, 2, 3, and 4 h) (Fig. 1d). DEX and INS exhibited opposite effects on *Per2* expression at 4 h, with DEX inducing and INS decreasing *Per2* mRNA (Fig. 1d). Notably, INS led to a fast reduction in PER2 protein after 1 h of treatment, while the levels were maintained for 4 h in the presence of glucose (Fig. 1c). Similarly, DEX and INS had converse impacts on *Cry1, Bmal1*, and *Dbp* mRNA expression. DEX diminished these clock genes' transcripts, whereas INS prompted their expression at 4 h (Fig. 1d). Furthermore, after 2 h of INS treatment, CRY1 protein levels declined; yet with glucose, INS induced increased CRY1 protein levels at 1 and 2 h (Fig. 1c). In summary, our findings indicate that acute changes in INS

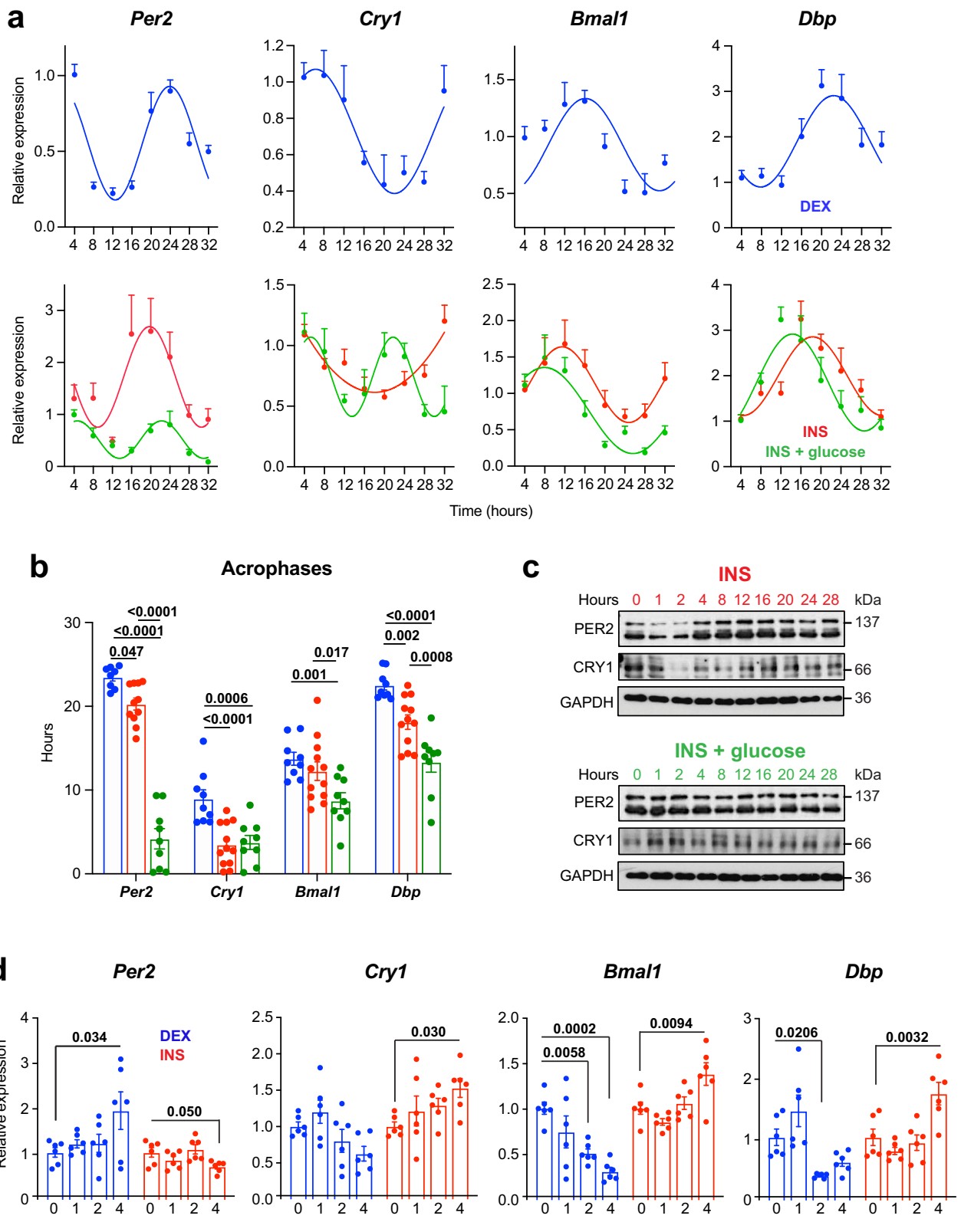

synchronize hypothalamic astrocyte clocks, and glucose's availability modulates the molecular clock's rhythmic expression. These observations suggest that INS may directly impact astrocyte entrainment and propose a secondary role for INS signaling in astrocytes, where changes in extracellular ligand concentration led to rhythmic expression of clock genes in this glial cell type.

**IR deletion in astrocytes impacts circadian locomotor activity in a sexually dimorphic fashion**

The SCN is insensitive to shifts in the feeding schedule[3] due to the solid inter-neuronal coupling[23]. However, IRs are distributed across the SCN in both neurons and astrocytes, with the former playing a pivotal role in central pacemaker intercellular coupling[25]. Furthermore, INS

**Fig. 1 | INS synchronizes the molecular clock in primary astrocytes. a** Primary hypothalamic astrocytes were synchronized with 100 nM of DEX (upper panel) or 600 nM INS in low or high glucose media (lower panel) for 1 h. *Per2, Cry1, Bmal1,* and *Dbp* expression at the indicated time points by quantitative RT-PCR. Graphs show the mean ± s.e.m. of the cosine-fitted curves from 3 (DEX and INS + glucose) or 4 (INS) experiments performed in triplicate. **b** Acrophase of the rhythmic oscillations of *Per2, Cry1, Bmal1,* and *Dbp* after the synchronization with DEX and INS. Graphs show the mean ± s.e.m. (DEX and INS + glucose: *n* = 3 experiments performed in triplicate, and INS: *n* = 4 experiments performed in triplicate). One-way ANOVA. **c** Representative images of PER2 and CRY1 western blots in hypothalamic astrocytes synchronized with INS in the absence or presence of high glucose (*n* = 2 independent experiments). **d** Primary hypothalamic astrocytes were synchronized with 100 nM of DEX or 600 nM INS and *Per2, Cry1, Bmal1,* and *Dbp* were analyzed, at the indicated time points, by quantitative RT-PCR. Data are represented as mean ± s.e.m (*n* = two experiments performed in triplicate). One-way ANOVA. Source data are provided as a Source Date file.

signaling has demonstrated its ability to reset the SCN clock[22], particularly influencing the lateral SCN region[22], recognized for its pacemaker robustness[53]. Thus, we expected that astrocytic IR signaling could influence SCN behavioral rhythms. To assess the role of INS signaling in astrocytes in vivo we utilized our established tamoxifen (TM)-inducible knockout mouse model[30] (GLASTCreERT2 +/−; IR[f/f], referred to as *IRcKO*), where the astrocyte-specific gene *Glast* promoter controls Cre-recombinase expression[54]. We administered TM to 10 weeks-old *IRcKO* male and female mice and littermate controls (*IR[f/f]*), an approach that we and others have previously validated for astrocyte- and time-specific gene deletion[24,25,29,30,54,55]. Using Rosa26 ACTB-tdTomato/EGFP (tdTomato/eGFP) reporter mice, we previously confirmed that Cre-mediated IR recombination occurred following TM treatment in males[30]. To investigate any potential sex-specific variations in recombination efficiency, we quantified the Tomato and GFP signals in the SCN and ARC of both male and female mice using these reporter mice (Supplementary Fig. 2a). Control mice exhibited around 15% Tomato-positive astrocytes in the SCN and 18% in the ARC, with no significant differences related to sex (Supplementary Fig. 2a). After TM treatment, both male and female mutants displayed a recombination percentage (denoted by GFP-positive astrocytes) of ~70% in the SCN and 60% in the ARC, without any significant sex-specific differences (Supplementary Fig. 2a). These findings indicate that both male and female mice exhibit comparable recombination rates in astrocytes within the SCN and ARC, as previously reported[29]. Consistent with these observations, we observed a reduction in IRβ in the hypothalamus of male *IRcKO* mice at *zeitgeber* time (ZT) 0 (Supplementary Fig. 2b). IR activates PI3K and mTOR complex 1 (mTORC1), which is a critical regulator of protein synthesis[56,57]. Hypothalamic phospho-mTOR levels were consistently reduced in *IRcKO* mice (Supplementary Fig. 2b). Furthermore, male mutants also exhibited decreased hypothalamic PER2 and CRY1 compared to controls, 2 months after TM treatment at ZT0 (Supplementary Fig. 2b). In female mice, there was a reduction in IRβ levels that did not reach statistical significance (Supplementary Fig. 2b). Like males, females exhibited a significant decrease in phospho-mTOR levels, as well as decreased PER2 expression and slightly diminished CRY1 expression at ZT0 (Supplementary Fig. 2b).

To uncover the contribution of astrocyte's INS signaling on the regulation of circadian locomotor activity, the wheel-running activity of control (*IR[f/f]*) and *IRcKO* animals was used as an index of SCN circadian function[58,59], 10 weeks following TM treatment. The animals were initially entrained to a 12–12 h light–dark (LD) cycle for a minimum of 7 days and then transitioned to constant darkness (DD) to observe their free-running rhythms. Subsequently, they were re-entrained to a new 12–12 h LD cycle (rLD)[25]. All *IRcKO* male mice aligned with expectations by entraining to the LD cycles and displayed comparable periodicity to their sex-matched controls (Fig. 2a, b and Supplementary Fig. 2c, d). In DD, both control and *IRcKO* male animals exhibited a similar daily rhythm with a period of ~24 h (Fig. 2b and Supplementary Fig. 2d). Recognizing the challenges associated with measuring re-entrainment following extended periods of DD, especially given the influence of individual variations in circadian periods, we observed that *IRcKO* mice adapted to the new LD cycles in a manner consistent with the control group (Fig. 2a, b and Supplementary

Fig. 2c, d). However, the mutant mice demonstrated a significant reduction in nocturnal activity compared to the control animals (Fig. 2a and Supplementary Fig. 2d). Conversely, among the female mutants, we observed decreased activity levels during the dark phase across all lighting conditions in comparison to control animals (Fig. 2a and Supplementary Fig. 2c, d). Notably, unlike their male counterparts, female *IRcKOs* displayed a reduction in the amplitude of the periodogram across LD, DD, and rLD, along with a significantly prolonged free-running period (Fig. 2b and supplementary Fig. 2d).

These findings indicate that INS signaling in astrocytes play a role in regulating light entrainment and dark phase activity patterns specifically in females, while its impact on male circadian locomotor activity appears to be limited. Previous research, encompassing both human and animal models, has underscored sex-related differences in circadian rhythms. Typically, females display heightened adaptability and resilience to phase shifts, whereas males tend to exhibit a more robust ability to sustain entrainment[29,60–62]. To investigate the potential impact of ovarian function on sex-related circadian variations in mutants, we performed bilateral ovariectomy (OVX) or sham operations on control and *IRcKO* female mice. Following a 4-week period to ensure the elimination of ovarian hormones[63–65], female mice subjected to sham operations or OVX were entrained to a 12–12 h LD cycle for a minimum of 8 days, and their locomotor activity was closely monitored. The results indicated that both sham operated and OVX female mutants exhibited reduced activity levels during the dark phase in comparison to control animals (Fig. 2c, d). Furthermore, they displayed a decrease in the amplitude of the LD periodogram (Fig. 2c). These findings suggest that the altered locomotor phenotype observed in *IRcKO* female mice does not depend on ovarian function. These results align with prior research indicating that estrogens are not obligatory for maintaining circadian rhythms[66]. While the impact of INS signaling could potentially extend beyond the SCN and influence locomotor activity by affecting circuits outside of it, our findings underscore the role of INS signaling in astrocytes in mediating sex-specific differences in circadian locomotor activity.

## Astrocytic IR regulation of feeding entrainment is sexually dimorphic in mice

As astrocytes are implicated in nutrient sensing[2], and INS signaling is involved in the entrainment to feeding time[22], we sought to determine whether the *zeitgeber* properties of restricted feeding (RF) would trigger changes in the food anticipatory activity (FAA) and the metabolic adaptation in *IRcKO* mice. Testing the former, 2 months after TM, *IRcKO* male and female mice were entrained by 12 h:12 h LD cycles (fed ad libitum), and then we gradually reduced food availability to 6 h for 6 days (ZT4-ZT10, RF 6 h), and finally to 4 h (ZT4-ZT8, RF 4 h) for 4 days (Fig. 3a upper panel). During 6 or 4 h of restricted feeding, FAA that began 1–4 h before food was provided as expected (Fig. 3b, d and Supplementary Fig. 3a) and was associated with a reduction of daily activity in the dark period in both controls and mutants (Supplementary Fig. 3b). However, the total activity remained similar between control and mutants (Supplementary Fig. 3c). These findings align with previous studies that have demonstrated how feeding time takes precedence over cues from the SCN under these conditions[67]. During

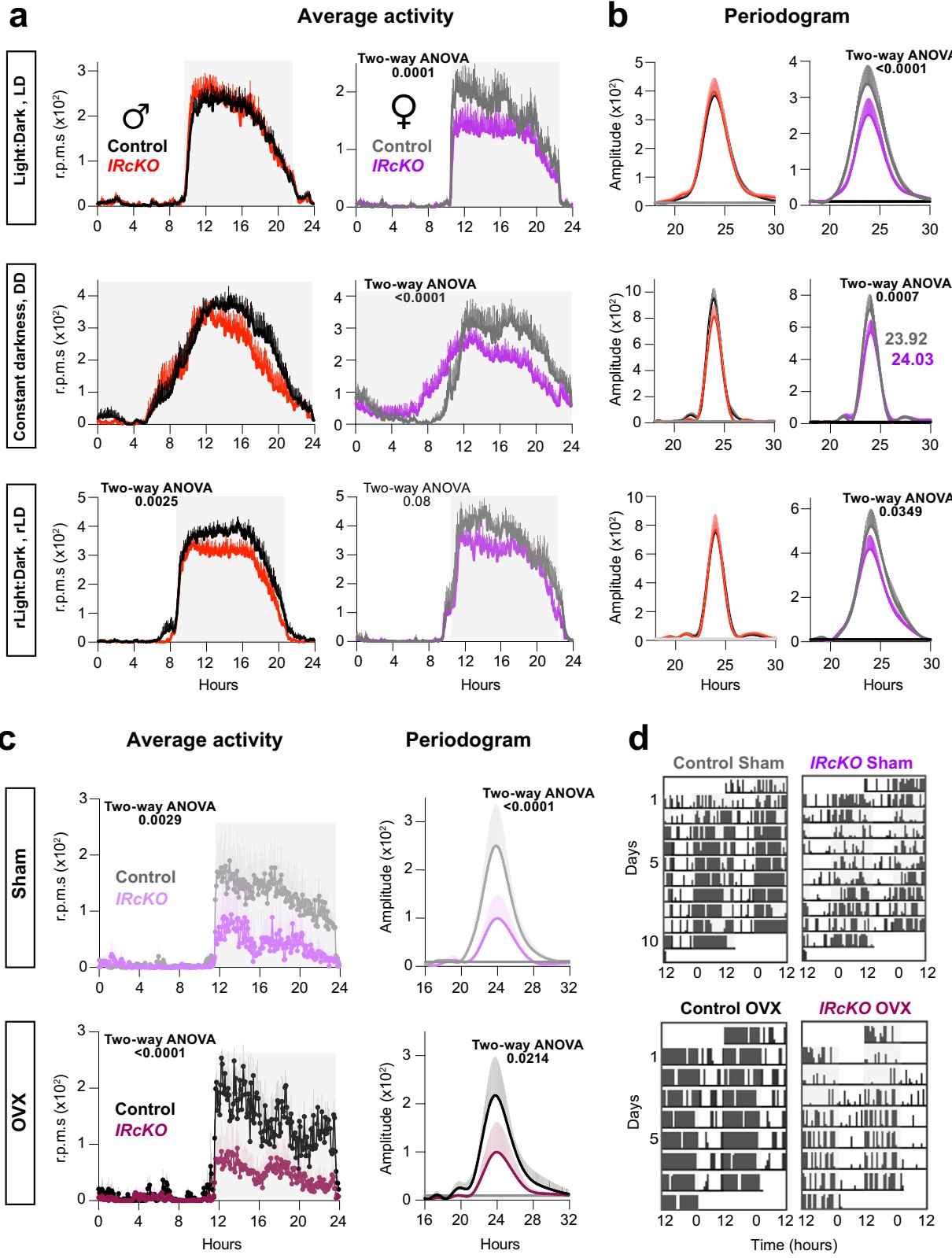

RF 6 h, mutants exhibited FAA patterns like those of controls (Fig. 3b, d). Interestingly, during the second phase of the paradigm (RF 4 h), female mutants displayed an intensified FAA occurrence 2 h prior to feeding compared to control animals (Fig. 3b, d). Thus, the deletion of IR in astrocytes did not impair the acquisition of the FAA. Indeed, the overall extent of FAA in *IRcKO* mice was equivalent to that observed in wild-type littermates (Supplementary Fig. 3a).

To further explore the role of astrocytic INS signaling in controlling behavioral anticipation, we conducted an experiment in which both control and *IRcKO* mice were subjected to reduced food availability for 4 h (ZT4-ZT8, RF4h) over 5 days to establish FAA. After FAA was established, we administered a combination of INS (2.25 U/Kg) and glucose (3 g/Kg)[22] at ZT12, through intraperitoneal injection, a time outside their regular mealtime during RF. Subsequently, the mice were

**Fig. 2 | IR deletion in astrocytes impacts circadian locomotor activity in a sexually dimorphic fashion. a** *IRcKO* and control mice were treated with TM, and 10 weeks after treatment, circadian locomotor activity was evaluated. Activity waveforms under the LD, DD, and rLD are shown for controls and *IRcKO* mice. Activity counts are averaged over 5-min intervals. LD and rLD data use nighttime hours 8–20, presented in ZT. DD employs circadian time (CT) on the *x*-axis, with mean activity as the average within 5-min bins across each animal's circadian cycle. Data are presented as mean ± s.e.m. (Males: $n = 27$–21 (LD), $n = 10$–9 (DD), and $n = 12$–9 (rLD) animals in control-mutant pairs. Females: $n = 8$–22 (LD), $n = 7$–13 (DD), and $n = 7$–12 (rLD) animals in control-mutant pairs). Two-way ANOVA. **b** Lomb–Scargle periodograms of control and *IRcKOs* in all lighting conditions. Female period in DD is indicated in the graphs. Data are presented as mean ± s.e.m.

(Males: $n = 20$–15 (LD), $n = 10$–9 (DD, rLD) animals in control-mutant pairs. Females: $n = 8$–21 (LD), $n = 7$–13 (DD), and $n = 7$–12 (rLD) animals in control-mutant pairs). Two-way ANOVA. **c**, **d** Control and *IRcKO* females underwent ovariectomy (OVX) or a sham procedure 3 weeks after TM treatment. Circadian locomotor activity was assessed 4 weeks post-surgery. **c** Activity waveforms (left panel, $n = 7$–4 (sham) and $n = 3$–5 (OVX) animals in control-mutant pairs) and Lomb–Scargle periodogram (right panel, $n = 6$–4 (sham) and $n = 3$–5 (OVX) animals in control-mutant pairs) under LD cycles for sham and OVX control and *IRcKO* females. Activity counts are averaged over 5-min bins, with data plotted from 8–20 nighttime hours and presented in ZT. Data are presented as mean ± s.e.m. Two-way ANOVA. **d** Representative actograms of sham and OVX control and *IRcKO* mice. Source data are provided as a Source Date file.

food-starved on the following day (Fig. 3a lower panel) to evaluate the phase-dependent effect of acute INS manipulation on FAA. This approach aimed to provide insights into the role of astrocytic INS signaling in the intrinsic circadian timing system that regulates FAA. Remarkably, our results indicate that INS and glucose can acutely shift the rhythm and duration of FAA, an effect that was absent in *IRcKO* male mice (Fig. 3b–d). Surprisingly, in females, INS and glucose did not impact the duration or phase of the FAA in both groups of mice (Fig. 3b, d). Overall, our results highlight that the deletion of IR in astrocytes did not modify the amplitude of food anticipatory rhythms. Nonetheless, acute variations in INS levels at specific phases of the food anticipation rhythm led to shifts in timing among males, suggesting a role of astrocytic INS signaling in determining the phase of FEOs in the brain. Furthermore, our results indicate that in males, INS plays a crucial role in communicating time-of-feeding to the astrocyte molecular clock, linking the periphery and the central clocks. Strikingly, the repercussions of IR loss specifically in male astrocytes disrupted the INS-induced phase shift in FAA, resulting in a FAA pattern like that observed in female mice. Thus, our results propose that astrocytic IR is a critical component of sex-specific circadian entrainment that regulates locomotor rhythms.

On the other hand, during the RF 6 h and RF 4 h paradigm, no differences in the cumulative food intake (Supplementary Fig. 3d) nor the body weight change (Supplementary Fig. 3e) were observed in the male mutants compared to control animals. However, *IRcKO* females gained less weight than controls during the RF paradigm (Supplementary Fig. 3e), consistent with their reduction in cumulative food intake (Supplementary Fig. 3d). Thus, these results indicate that while astrocyte ablation of IR in females is not required to anticipate daily meals, it is sufficient to impair metabolic adaptation to RF.

**Dopamine receptor 2 agonism rescue the circadian behavior of *IRcKO* mice**

It has been reported that the deletion of IR on astrocytes within the nucleus accumbens of the striatum leads to a decrease in ATP exocytosis, resulting in reduced purinergic signaling on dopaminergic neurons. As a consequence, there is a decrease in dopamine (DA) release, which ultimately contributes to the development of increased anxiety- and depressive-like behaviors in mice[31]. Notably, DA plays a crucial role in feeding, locomotor behavior, as well as photic and food entrainment[68]. Indeed, studies have demonstrated that DA antagonists and DRD1 knockout lead to diminished FAA[8], while a DRD2 agonist can alter FAA, and DRD2 antagonists decrease total activity before scheduled mealtimes[69,70]. Hence, it is plausible that reduced DA signaling in *IRcKO* mice might underlie the altered circadian locomotor activity and impaired INS-induced FAA phase shift. To investigate this hypothesis, we analyzed the levels of DA and its metabolites (3,4-dihydroxyphenylacetic acid–DOPAC and homovanillic acid–HVA), along with noradrenaline (NA) and serotonin (5-HT) and its metabolite 5-hydroxyindolacetic acid (5-HIAA) in the striatum and hypothalamus of control and *IRcKO* male mice[71–74]. Our findings revealed reduced levels of DOPAC in the striatum of the mutants, resulting in diminished

ratios of DOPAC/DA and DOPAC + HVA/DA (Fig. 4a). However, the levels of DA, NA, 5-HT, and 5-HIAA remained unchanged. These results indicate that the observed changes in dopaminergic signaling are more likely attributed to alterations in DA release (as evidenced by reduced DOPAC levels) rather than changes in their overall availability or synthesis. In the hypothalamus, we noted significantly elevated levels of DA, DOPAC, and NA (Fig. 4b), along with an increased DOPAC/DA ratio (Fig. 4b), indicating an augmented synthesis of catecholamines. These findings emphasize the regional specificity of the modified DA turnover in *IRcKO* mice, with the hypothalamus exhibiting heightened DA synthesis and turnover, while the striatum displays disruptions in DA release. Therefore, the altered DA turnover in *IRcKO* mice does not indicate a general defect in neurosecretory granule release, aligning with previous findings in brain-specific IR knockout mice[75].

Expanding upon these findings, our aim was to further investigate the role of DA signaling in the altered circadian behaviors observed in *IRcKO* mice. For this purpose, we administered the DRD2 agonist quinpirole (1 mg/kg per day via intraperitoneal injection) or a vehicle to both control and *IRcKO* mice at ZT12 for a duration of 5 days. Throughout this time, we closely monitored their daily locomotor activity patterns. Following the treatment, we observed a significant decrease in nocturnal activity for both control and male mutant mice (Fig. 4c and Supplementary Fig. 4a), along with a notable suppression of the amplitude of the periodogram (Fig. 4d). While we did observe a slight decrease in periodicity for both groups, it did not reach statistical significance (Supplementary Fig. 4b). To further investigate the role of dopaminergic signaling in controlling the FEOs that drive behavioral anticipation in males, once the FAA of control and *IRcKO* mice was established, after mice were subjected to reduced food availability to 4 h (ZT4-ZT8, RF 4 h) for 5 days, mice were treated with DRD2 agonist quinpirole or vehicle at ZT12, and food starved the next day. Importantly, treatment with the DRD2 agonist induced a shift in the rhythm and duration of FAA in both control and mutant animals (Fig. 4e and Supplementary Fig. 4c). This shift was comparable to the shift induced by INS in control animals but was significantly impaired in the mutant mice (Fig. 3b, d).

DRD2 agonists have exhibited varying effects on locomotion between males and females, with a more pronounced reduction observed in males compared to females[76,77]. In line with this, we found that the administration of a DRD2 agonist actually increased nocturnal activity in females (Fig. 4f). Interestingly, this effect was due to a delayed onset of activity induced by the DRD2 agonist, resulting in a shorter active period. Notably, this adjustment failed to induce changes in total locomotor activity (Fig. 4i). Similar to males, a significant reduction in the periodogram was detected in both control and *IRcKO* females (Fig. 4g, h). However, our observations extended beyond the treatment period. Continuing to monitor the locomotor activity of control and *IRcKO* mice after withdrawal of the DRD2 agonist for 7 days revealed a remarkable development. Notably, the locomotor activity of both control and mutant mice increased post-drug withdrawal (Fig. 4f). This increase was particularly evident in the mutants, which, after 7 days, exhibited similar nocturnal activity to the control group

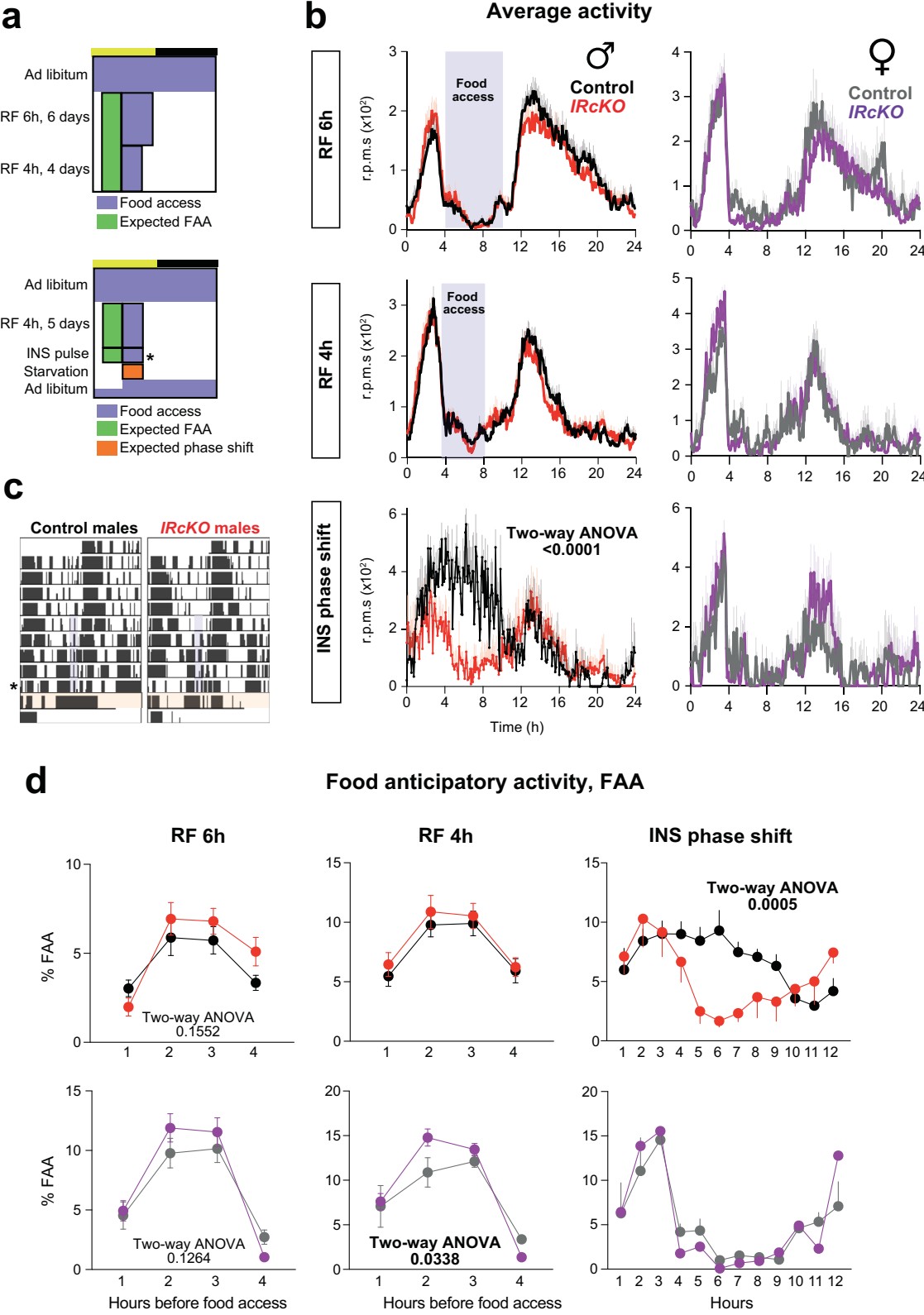

(Fig. 4f, i). Furthermore, the amplitude of the periodogram increased in both groups (Fig. 4g, h). Additionally, a slightly longer period was observed in control animals after drug withdrawal, whereas this lengthening of the period was particularly evident in the mutants (Supplementary Fig. 4d). In sum, even though the DRD2 agonist also affected control mice, the short-term administration of the DRD2

agonist led to a sustained (1-week) normalization of locomotor activity in female mutants.

In summary, our findings indicate disrupted dopaminergic system resulting from the loss of INS signaling in astrocytes affects the INS-induced entrainment of the FEO in males and the light entrainment in females. Importantly, the administration of the DRD2 agonist restores

**Fig. 3 | Astrocytic IR regulation of entrainment to feeding is sexually dimorphic in mice. a** Diagrams showing the food restriction paradigms. **b** Average waveforms of control and *IRcKO* mice under food restriction (RF) from ZT 4–10 (RF 6 h), ZT 4–8 (RF 4 h), and after the INS and glucose bolus administration. Purple areas in the graphs indicate the feeding time. Data are presented as mean ± s.e.m. (Males: *n* = 24–19 (RF6h), *n* = 29–24 (RF4h), *n* = 5 (phase shift) animals in control-mutant pairs. Females: *n* = 5–8 animals in control-mutant pairs). Two-way ANOVA. **c** Representative actograms depicting the circadian behavior of control and *IRcKO*

mice ad libitum, during the RF 4 h and the phase shift induced by the administration of INS and glucose. Purple areas indicate the feeding time; the asterisk indicates the day mice received the bolus of INS and glucose (at ZT12); orange areas show the expected phase shift after the administration of INS and glucose. **d** Percentage of the hourly FAA in control and *IRcKO* mice. Data are presented as mean ± s.e.m. (Males: *n* = 24–20 (RF6h), *n* = 29–25 (RF4h), *n* = 5 (phase shift) animals in control-mutant pairs. Females: *n* = 5–8 animals in control-mutant pairs). Two-way ANOVA. Source data are provided as a Source Date file.

the phase shift of the FAA induced by INS in *IRcKO* males and the nocturnal activity in *IRcKO* females, respectively.

## Astrocytic IR regulation of energy balance is sexually dimorphic in mice

The disruption of the circadian cycle, observed in individuals working night shifts or experiencing circadian arrhythmia in rodent models, closely links to metabolic imbalance and INS resistance[2,78,79]. Intriguingly, processes governed by circadian rhythms, including metabolism, feeding patterns, sleep-wake cycles, hormone secretion, and body temperature regulation, exhibit sex-related differences[62]. Recent insights have highlighted the influence of astrocyte circadian rhythms on energy balance and INS sensitivity, with a noted sex dependency[24,29]. Despite these observations, the precise regulatory circuits and signaling mechanisms underlying these sex-based differences remain poorly understood. Given the intricate interplay between circadian rhythms, the sex-specific variations in metabolic regulation, and the notable sexual dimorphism observed in the circadian behavior of *IRcKOs*, we sought to delve deeper into understanding the metabolic phenotype of *IRcKO* mice in both males and females. We found that, compared to their sex-matched controls, both male and female mutants had lower body weight (week 8 for males and week 3 for females) after treatment with TM (Fig. 5a). This reduction in body weight was accompanied by significantly decreased adiposity, including lower white adipose tissue (WAT) depots in the gonadal, subcutaneous, and visceral regions, as well as reduced lean mass after 20 weeks of TM treatment (Fig. 5b). As astrocytic INS signaling impacted fat mass in both males and females, we analyzed serum cholesterol, triglycerides, and leptin levels in both groups of animals 20 weeks after TM treatment. We found significantly lower serum levels of triglycerides, cholesterol, and leptin in *IRcKO* male mice compared to control animals (Supplementary Fig. 5a). Conversely, female mutants had significantly higher serum triglyceride levels compared to control animals, and while there was a slight increase in serum cholesterol, it did not reach statistical significance (paired *t*-test, *p* = 0.181). Additionally, their leptin levels were comparable to controls (Supplementary Fig. 5a). While ablation of astrocytic IR impairs glucose metabolism in males[30], we found that *IRcKO* females showed significantly improved glucose tolerance (Supplementary Fig. 5b). The reduced body weight of the male mutants was not due to hypophagia (Fig. 5c), or alterations in either brown adipose tissue (BAT) thermogenesis (Supplementary Fig. 5c), or daily spontaneous locomotor activity (Fig. 5e). Instead, it was likely a result of increased energy expenditure during the light and dark phases (Fig. 5f). On the other hand, the negative energy balance of female *IRcKOs* could be attributed to their reduced food intake (Fig. 5c) and increased energy expenditure (Fig. 5e), which compensated for their reduced daily activity (Fig. 5d) and normothermia (Supplementary Fig. 5c). ANCOVA analysis of calorimetric data showed that energy expenditure adjusted for body mass was not affected by astrocytic IR deficiency in males. However, in females, the absence of astrocytic IR signaling impacted the relationship between body weight and energy expenditure, leading to a decrease in energy expenditure as body weight increased (Fig. 5e). Additionally, the deletion of IRs in female astrocytes resulted in a decrease in the respiratory quotient during both the day and night

(Fig. 5f), suggesting a better regulation of energy balance and lower adiposity through decreased carbohydrate oxidation. This shift in energy utilization, which was observed only in females, may reflect a differential sensitivity to the catabolic effects of INS based on sex. Altogether, our results indicate that INS signaling in astrocytes leads to a negative energy balance in both male and female mice by regulating feeding, energy expenditure, and spontaneous locomotor activity in a sex-dimorphic manner.

To investigate the potential involvement of ovarian function in the observed sex differences in metabolic dysregulation in the mutants, control or *IRcKO* females underwent bilateral OVX or sham-operated procedures, as previously described[29,63,64]. The surgeries were performed after 3 weeks of TM treatment when female IR-deficient mice began showing decreased body weight (Fig. 5g). OVX controls and mutants exhibited similar weight gain after 3 weeks of the procedure (Fig. 5g) and showed comparable increases in adiposity (Fig. 5h) and serum leptin levels (Supplementary Fig. 5d). OVX *IRcKOs* normalized their food intake (Supplementary Fig. 5e). Moreover, the glucose tolerance test revealed that OVX worsened glucose tolerance similarly in both control and *IRcKO* females (Supplementary Fig. 5f). Interestingly, while energy expenditure was reduced after OVX in both control and mutant mice, mutants still showed significantly increased levels (Fig. 5i). These findings suggest that the differences in body weight and composition, food intake, and glucose homeostasis observed in IR-deficient mice compared to controls are influenced by ovarian function.

## Astrocytic IR differentially regulates energy balance in diet-induced obesity (DIO)

Disruption of the normal circadian cycle can result from nutritional challenges, such as the consumption of a high-fat diet (HFD)[80]. These challenges initiate significant reorganization in specific metabolic pathways, thereby remodeling peripheral clocks[80]. Importantly, these alterations primarily stem from the nutritional challenges themselves rather than diet-induced obesity (DIO)[80]. To further understand the intricate role of astrocytic INS signaling in governing energy balance and circadian rhythms, we investigated the effects of a 60% high-fat diet (HFD) over a 20-week period in male and female *IRcKO* mice. Male *IRcKO* mice consistently showed reductions in body weight and fat mass (Fig. 6a), mirroring observations in mice on a standard diet (STD) (Fig. 5a, b). Interestingly, despite elevated levels of serum insulin-like growth factor 1 (IGF1) (Supplementary Fig. 6a) male *IRcKOs* exhibited impaired growth (Fig. 6b), indicating that the loss of astrocytic IR exerts a dominant effect on growth regulation, overriding the potential compensatory actions of elevated IGF1 signaling[81]. It's noteworthy that the differences observed in serum triglycerides and cholesterol between male *IRcKO* mice and controls on the STD (Supplementary Fig. 5a) were no longer present when the mice were challenged with HFD (Supplementary Fig. 6a). Nevertheless, male *IRcKO* mice on the HFD continued to exhibit reduced fat mass (Fig. 6a) and lower serum leptin levels (Supplementary Fig. 6a), suggesting that astrocytic INS signaling continues to impact adipose tissue and leptin secretion during HFD conditions. Furthermore, male *IRcKOs* displayed comparable food intake to controls on the HFD (Supplementary Fig. 6b) and showed a trend of improved glucose homeostasis, though not

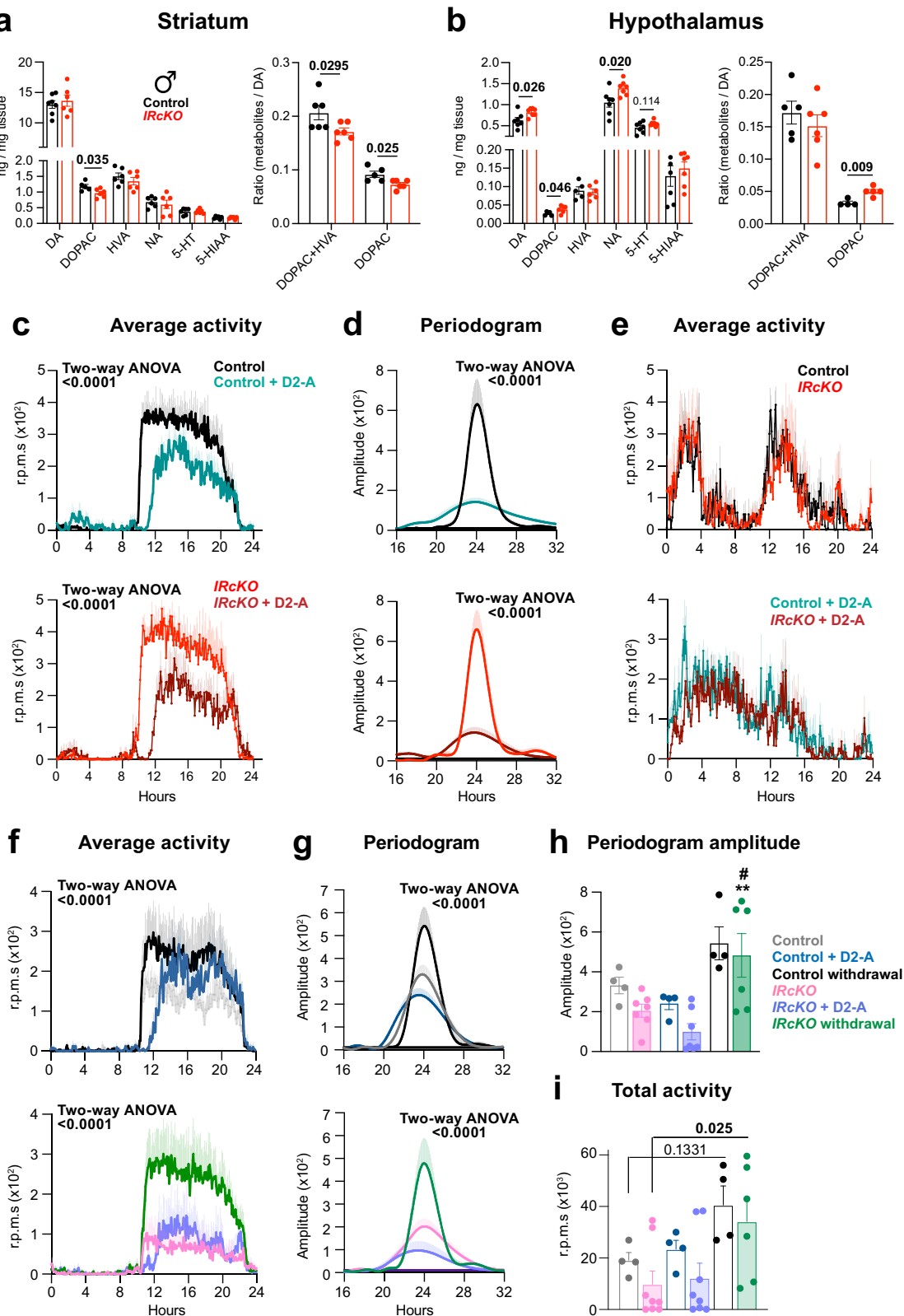

reaching statistical significance (Supplementary Fig. 6c). Importantly, HFD-fed mutants exhibited increased rectal and BAT temperatures (Fig. 6c), as well as elevated uncoupling protein 1 (UCP1) expression in BAT (Fig. 6d). This suggests a significant role of astrocytic INS signaling in modulating the thermogenic response to an HFD in males. Notably, male *IRcKO* mice on the HFD exhibited increased energy expenditure (Fig. 6e) with respiratory quotients comparable to controls

(Supplementary Fig. 6d). In summary, male *IRcKO* mice displayed resistance to DIO, likely due to increased thermogenesis and higher energy expenditure, both intricately regulated by astrocytic INS signaling.

Given the observed resistance of *IRcKO* mice to DIO, we analyzed hypothalamic INS signaling and clock protein expression in these HFD-challenged mutants at ZT0 and ZT12. We observed increased levels of

**Fig. 4 | Impact of the dopamine receptor 2 agonism on the circadian locomotor behavior of *IRcKO* mice.** Levels and ratios of DA, DOPAC, HVA, NA, 5-HT and 5-HIAA in the striatum ($n = 6$ mice per group) (**a**) and hypothalamus ($n = 4$ and 5 mice) (**b**) of control and *IRcKO* males. Data are presented as mean ± s.e.m. Two-tailed, unpaired *t*-test. **c** Activity waveforms and (**d**) Lomb–Scargle periodograms of control ($n = 6$) and *IRcKO* ($n = 5$ and 4) male mice treated with the DRD2 agonist quinpirole (1 mg/kg per day) or vehicle for 5 days. Activity counts are expressed as the average amount of activity in 5-min bins, and data are plotted with nighttime hours from 8–20 and given in ZT. Two-way ANOVA. **e** Average waveforms of control and *IRcKO* male mice under food restriction for 5 days ($n = 7$ and 5) and after the DRD2 agonist treatment ($n = 6$ animals per group) at ZT12. **f** Activity waveforms of control and *IRcKO* females treated the DRD2 agonist quinpirole (1 mg/kg per day)

for 5 days and after the drug withdrawal (7 days). Activity counts are expressed as the average amount of activity in 5-min bins, and data are plotted with nighttime hours from 8 to 20 and given in ZT ($n = 4$ and 5 for controls; $n = 8$ and 6 for mutants). Two-way ANOVA. **g** Periodogram of control and *IRcKO* females treated the DRD2 agonist quinpirole (1 mg/kg per day) for 5 days and after the drug withdrawal (7 days). ($n = 4$ for controls; $n = 7$ and 6 for mutants). Two-way ANOVA. **h** Amplitude of the Lomb–Scargle periodograms and (**i**) total activity of female control ($n = 4$) and *IRcKOs* ($n = 7$ and 6) treated with a DRD2 agonist and after the withdrawal of the drug. Two-way ANOVA (# = 0.0284 versus *IRcKO* mice; ** = 0.0013 versus *IRcKO* + DRD2 agonist). Data are represented as mean ± s.e.m. Source data are provided as a Source Date file.

IRβ and pAKT at ZT0, and elevated GLUT1 levels at both ZT0 and ZT12 (Supplementary Fig. 6e). These findings suggest an upregulation of INS signaling and glucose transport in the hypothalamus of HFD-challenged mutants. Notably, DIO-challenged *IRcKOs* exhibited heightened levels of hypothalamic PER2 and CRY1 (Supplementary Fig. 6e), contrasting the reductions observed in STD conditions (Supplementary Fig. 2b). This suggests a differential clock-related response to diet challenges, where DIO prompts an unexpected rise in these clock proteins among *IRcKOs*. The elevated hypothalamic INS signaling in mutants may enhance thermogenesis and improve metabolic impairment, potentially explaining the negative energy balance and obesity protection observed in male *IRcKO* mice on the HFD. Together, these results highlight the complex interplay between astrocytic INS signaling, clock protein expression, and energy homeostasis during DIO.

When subjecting female mutants to an HFD, intriguing shifts in their metabolic profile became evident. Specifically, they displayed reduced body weight, adiposity, and lean mass (Fig. 6f). Despite these alterations, no significant variations in serum levels of leptin, triglycerides, or cholesterol were observed between the diet-induced obesity (DIO)-challenged *IRcKO* mice and control animals (Supplementary Fig. 6f). Interestingly, female mutants showed a remarkable reduction in food intake (Fig. 6g) suggesting potential shifts in appetite regulation. Furthermore, despite the HFD challenge, *IRcKO* females maintained improved glucose tolerance compared to control mice (Supplementary Fig. 6g), mirroring the findings observed in mice on a STD (Supplementary Fig. 5b). In addition to these observations, DIO-challenged *IRcKO* females demonstrated heightened spontaneous activity (Fig. 6h), particularly during the daytime. This suggests that the HFD effectively reversed the previously noted reduced locomotor activity among female mutants under STD conditions (Fig. 5d). While female *IRcKOs* exhibited normal thermogenesis (Fig. 6i), they showed increased energy expenditure upon HFD consumption (Fig. 6j), which might contribute to the observed reduced body weight and fat mass in HFD-challenged female *IRcKO* mice. Importantly, we also noted a lower respiratory quotient in the mutants (Fig. 6k), indicating that HFD-fed *IRcKO* females relied more on lipids as their primary energy source compared to controls.

In summary, these findings collectively highlight the pivotal role of astrocytic INS signaling in maintaining a negative energy balance in both male and female mice. This effect is achieved through the intricate regulation of food intake, BAT thermogenesis, locomotor activity, and energy expenditure in mice challenged with an HFD. Importantly, these findings highlight the sex-dependent effects of astrocytic INS signaling on energy balance, revealing distinctive metabolic responses in males and females.

Given the impaired acquisition of FAA caused by a HFD[82,83] and the observed resistance to DIO in *IRcKO* mice, we explored the effects of timed RF on FAA acquisition and metabolic adaptation in male mutants. Remarkably, male *IRcKO* mice on the HFD exhibited a significant increase in nocturnal activity and amplitude of the periodogram compared to controls (Supplementary Fig. 6h, i). During the RF

paradigm on the HFD, control mice displayed reduced FAA (Supplementary Fig. 6h, i), consistent with previous studies[82,83]. Intriguingly, male *IRcKO* mice exhibited elevated FAA compared to controls, particularly at RF4h (Supplementary Fig. 6i). Notably, the mutants also demonstrated a more substantial decrease in the percentage of nocturnal activity during RF, while their overall activity remained unchanged (Supplementary Fig. 6j). Importantly, feeding schedules and caloric restriction have known effects on the phase of nocturnal activity and light-entrained rhythms[67,84]. In line with these observations, control animals experienced a phase advancement in nocturnal activity during RF, while the mutants displayed a more pronounced phase shift, especially at RF4h (Supplementary Fig. 6j). These results suggest that (1) astrocytic INS signaling might play a role in suppressing phase shifts in response to RF and (2) contribute to the adverse effects of HFD on FAA. Notably, male *IRcKO* mice demonstrated less weight loss compared to controls during the RF paradigm on the HFD (Supplementary Fig. 6k), likely attributed to their higher cumulative food intake (Supplementary Fig. 6k). This elevated food intake could potentially serve as a compensatory response to the restricted food availability during RF, aiming to meet their heightened energy demands.

In conclusion, these findings collectively underscore the pivotal role of astrocytic INS signaling in regulating FAA and facilitating metabolic adaptation, particularly in the context of dietary challenges like RF and HFD.

## Hypothalamic astrocytic IR differentially regulate circadian behavior and energy balance in males and females

The mediobasal hypothalamus (MBH), encompasses the ARC and the ventromedial nucleus of the hypothalamus (VMH)[2,85,86]. The VMH integrates information from light exposure and nutrient availability, while the ARC contains an SCN-independent oscillator that is sensitive to feeding states[85,87]. Despite the importance of the MBH in coordinating metabolic and circadian processes, the role of MBH astrocytes' INS signaling in sustaining circadian entrainment and regulating clock outputs, such as locomotor activity and daily energy homeostasis, remains poorly understood. To address this gap in knowledge, we utilized a viral-mediated Cre/lox system approach to selectively delete IR exclusively in astrocytes located in the MBH of adult male or female *IR^{fl/fl}* mice, as we have previously reported[30]. Our previous study demonstrated the efficacy of this viral approach in knocking out IRβ[30]. In the present study, we confirmed the successful targeting of MBH astrocytes through GFP labeling, indicating specific infection within the MBH (Fig. 7a). Additionally, these mice exhibited a significant 32% decrease in IRβ protein levels in the MBH (Fig. 7a). Upon deleting astrocytic IR in the MBH (IR-MBH-KO) in males, we observed alterations in the metabolic phenotype that did not replicate that of *IRcKO* mutants. Specifically, IR-MBH-KO mice did not show significant changes in body weight or composition (Supplementary Fig. 7a) unlike the observations in *IRcKOs*. However, mutants displayed a significant reduction in cumulative food intake (Fig. 7b) and energy expenditure (Fig. 7c), which contrasted with the unchanged or increased values

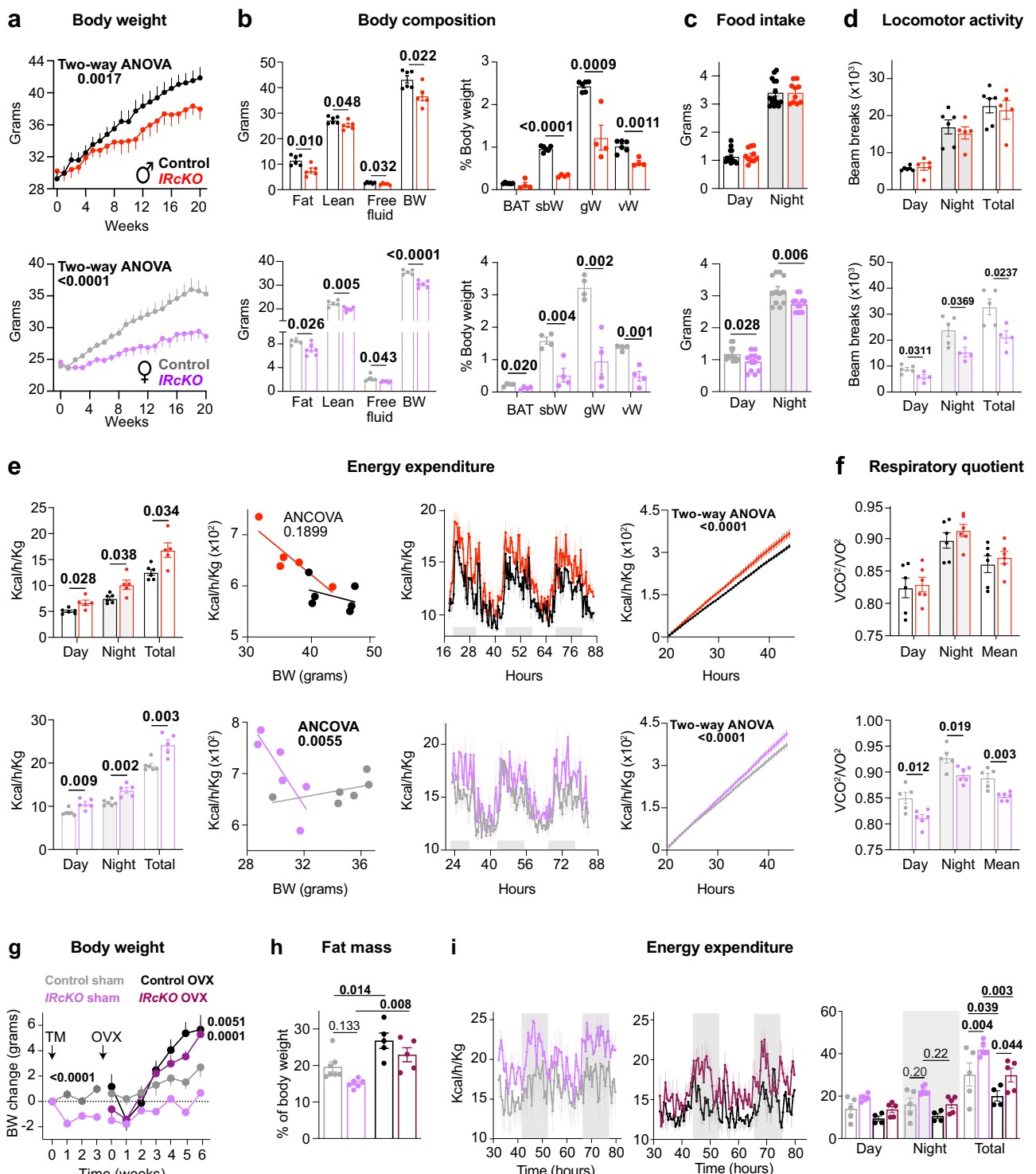

**Fig. 5 | Astrocytic IR regulation of energy balance is sexually dimorphic in mice.**
**a** Age-dependent changes in body weight of control ($n = 15$ males and $n = 12$ females) and *IRcKO* mice ($n = 10$ males and $n = 12$ females). Two-way ANOVA. **b** Left, panel: body composition of control ($n = 6$ males and $n = 5$ females) and *IRcKO* ($n = 5$ males and $n = 6$ females) mice (BW, body weight). Right panel: weights of brown adipose tissue (BAT), subcutaneous, gonadal, and visceral white adipose tissues (scWAT, gWAT, and vWAT) of control ($n = 6$ males and $n = 4$ females) and mutants ($n = 4$ animals per group). Two-tailed, unpaired *t*-test. **c** Food intake of control ($n = 14$ males and $n = 12$ females) and *IRcKO* ($n = 10$ males and $n = 12$ females) mice 21 weeks after TM treatment. Two-tailed, unpaired *t*-test. **d** Total spontaneous activity of control ($n = 6$ males and $n = 5$ females) and *IRcKO* ($n = 5$ males and $n = 4$ females) mice at 25 weeks after TM treatment in regular LD cycles. Two-tailed, unpaired *t*-test. **e** Hourly plots show hourly energy expenditure and bar plots show

total energy expenditure in control ($n = 6$) and *IRcKO* mice ($n = 5$). Two-way ANOVA or two-tailed, unpaired *t*-test. **f** Respiratory quotient during the light and dark phase in control and *IRcKO* mice (controls: $n = 6$ males and $n = 5$ female; mutants: $n = 6$ animals per group). Two-tailed, unpaired *t*-test. **g** Control and *IRcKO* females were subjected to ovariectomy (OVX) or sham procedure 3 weeks after TM treatment. Age-dependent changes in the body weight of sham and OVX control ($n = 3$ and $n = 5$) and *IRcKO* females ($n = 3$ and 4). Two-way ANOVA. **h** Fat mass of sham and OVX control ($n = 7$ and 5) and *IRcKO* ($n = 6$ and n = 5) females, 6 weeks after surgery. Two-way ANOVA. **i** Hourly plots show hourly energy expenditure and bar plots show total energy expenditure in sham or OVX control ($n = 5$ and 4) and *IRcKO* ($n = 6$ and 5) mice. Two-way ANOVA. Data are represented as mean ± s.e.m. Source data are provided as a Source Date file.

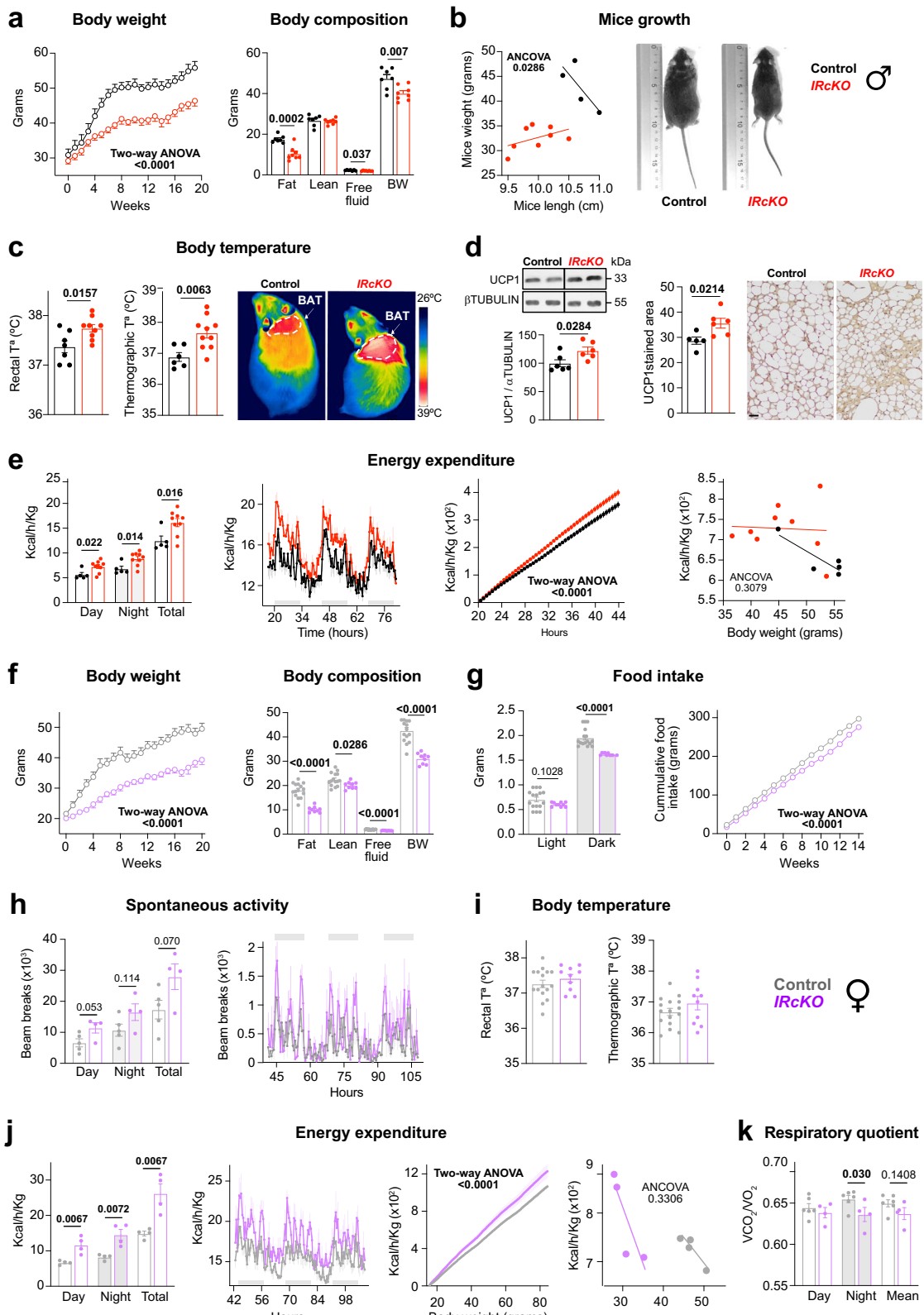

observed in *IRcKO* males. The respiratory quotient of IR-MBH-KD males was largely consistent with that of control animals (Supplementary Fig. 7b), resembling the findings in *IRcKO* males. Notably, while the locomotor activity of IR-MBH-KD males was comparable to that of control animals (Fig. 7d), they exhibited an increased amplitude of the periodogram (Fig. 7e). Despite *IRcKO* males displaying impaired shifts in the FAA following acute INS level manipulations, our further

investigation into whether INS signaling in mediobasal astrocytes could trigger changes in the *zeitgeber* properties of RF did not yield significant alterations. Specifically, the deletion of IR in MBH astrocytes did not alter the FAA during the RF 6 h and RF 4 h paradigms, nor did it affect the INS-induced phase shift (Supplementary Fig. 7c). These results suggest that while mediobasal astrocytic INS signaling plays a role in controlling food intake and energy expenditure, other brain

**Fig. 6 | Astrocytic IR differentially regulates energy balance in diet-induced obesity (DIO).** Left panel: age-dependent changes in body weight of control and *IRcKO* males (*n* = 7 and 8) (**a**) and females (*n* = 7) (**f**) challenged with a HFD starting 1 week after TM treatment. Right panel: body composition of control and *IRcKO* males (*n* = 7 and 8) (**a**) and females (*n* = 15 and 9) (**f**) after 12 weeks of HFD (BW, body weight). Two-way ANOVA or two-tailed, unpaired *t*-test. **b** Length of DIO-challenged control (*n* = 4) and *IRcKO* (*n* = 8) males after 14 weeks of HFD and representative image of a control and mutant mice. ANCOVA. **c** Rectal, BAT thermographic temperature and representative infrared thermal images in DIO-challenged control (*n* = 7 and 6) and *IRcKO* (*n* = 9 and 10) males after 9 weeks of HFD. Two-tailed, unpaired *t*-test. **d** BAT UPC1 protein levels (*n* = 6), quantification, and representative pictures of BAT immunohistochemistry for UPC1 (*n* = 5 and 6) in DIO-challenged control and *IRcKO* males. Scale bar 10 µm. Two-tailed, unpaired *t*-test. **e** Hourly plots show hourly energy expenditure and bar plots show total energy expenditure

in controls (*n* = 5) and *IRcKO* (*n* = 9) mice challenged with HFD for 12 weeks. Two-tailed, unpaired *t*-test or two-way ANOVA. **g** Daily and cumulative food intake of DIO-challenged control (*n* = 16 and 13) and *IRcKO* (*n* = 10 and 5) females. Two-tailed, unpaired *t*-test or two-way ANOVA. **h** Hourly plots show hourly spontaneous activity and bar plots show total spontaneous activity in control (*n* = 5) and *IRcKO* (*n* = 4) mice challenged with HFD. Two-tailed, unpaired *t*-test. **i** Rectal, BAT thermographic temperature and representative infrared thermal images in DIO-challenged control (*n* = 16) and *IRcKO* (*n* = 10) females after 9 weeks of HFD. **j** Hourly plots show hourly energy expenditure and bar plots show total energy expenditure in controls and *IRcKO* female challenged with HFD (*n* = 4). Two-tailed, unpaired *t*-test or two-way ANOVA. **k** Respiratory quotient of DIO-challenged control (*n* = 6) and *IRcKO* (*n* = 4) female mice. Two-tailed, unpaired *t*-test. Data are represented as mean ± s.e.m. Source data are provided as a Source Date file.

regions and circuits likely contribute to the broader metabolic and locomotor phenotype observed in *IRcKO* males.

On the other hand, female IR-MBH-KO mice did not show significant alterations in their body weight or composition, cumulative food intake, energy expenditure, or respiratory quotients (Supplementary Fig. 7d–g), as observed in *IRcKO* mice. Notably, the deletion of astrocytic IR in female mediobasal astrocytes did lead to a suppression of nocturnal activity (Fig. 7d) and a reduction in the amplitude of the periodogram in LD cycles (Fig. 7e). This phenotype closely resembled that of *IRcKO* mice. These results suggest that INS signaling in female MBH astrocytes is likely involved in the locomotor phenotype of *IRcKO* mutants. Overall, the findings suggest that INS signaling in MBH astrocytes regulates food intake, energy expenditure, and locomotor activity, but, of note, the specific effects differ between male and females.

To investigate the specific roles of hypothalamic astrocytes further, we utilized a viral-mediated Cre/lox system to selectively knock out IR in astrocytes of the VMH (IR-VMH-KD) and ARC (IR-ARC-KD). We confirmed the successful targeting of VMH or ARC astrocytes through GFP labeling, demonstrating specific infection within those hypothalamic nuclei (Fig. 7f and Supplementary Fig. 8a). Moreover, we demonstrated that ARC-mutants exhibited a significant decrease in IRβ protein levels in the ARC (Fig. 7f). In both male and female IR-VMH-KO mice, there were no significant differences in either body weight, body composition, or food intake compared to control mice (Supplementary Fig. 8b, c). The locomotor activity of male and female mutants was largely preserved, while female *IRcKOs* exhibited a surprising increase in the amplitude of the periodogram (Supplementary Fig. 8d), contrasting with the phenotype observed in *IRcKO* mice, characterized by reduced locomotor activity and periodogram amplitude. These findings suggest that while the deletion of astrocytic IRs within the VMH does not account for the metabolic and circadian phenotypes observed in male *IRcKO* mice, it implies a potential role in regulating the circadian locomotor activity in females.

In male IR-ARC-KD mice, no significant alterations were observed in body weight, body composition, food intake, or locomotor activity compared to control animals (Supplementary Fig. 8e–g). Notably, in female IR-ARC-KD mice, a different pattern emerged, resembling the metabolic alterations observed in *IRcKO* females. Specifically, IR-ARC-KD females exhibited reduced body weight, cumulative food intake, and locomotor activity, along with increased energy expenditure and reduced respiratory quotient compared to control mice (Fig. 7g–k). These results suggests that astrocytic IR signaling in the ARC may play a significant role in regulating energy balance and daily locomotor activity specifically in females. The contrasting effects between male and female IR-ARC-KD mice further highlight the complexity of astrocytic INS signaling in the hypothalamus and its sex-specific roles in metabolic and circadian regulation.

Overall, our findings suggest that in males, brain regions and circuits other than the MBH are responsible for the metabolic and

circadian phenotypes, while in females, INS signaling in ARC astrocytes is crucial for the metabolic and circadian phenotypes. This underscores the sex-specific roles of hypothalamic astrocytic INS signaling in regulating metabolic and circadian processes.

## Discussion

This study demonstrates a role for astrocytic INS signaling in light and food entrainment in mice. Importantly, pharmacological modulation of DRD2-receptor signaling partially rescued the circadian phenotypes of *IRcKO* mutants, demonstrating a physiological regulation of neuronal function rather than a neural degeneration-induced phenotype.

The FEO is a master multioscillatory DA-responsive circadian clock system for food anticipation[8]. It was recently proposed that INS signaling triggered after feeding is necessary and sufficient for both the FAA and the phase-shift of body clocks by regulating PER2[22]. Our results, along with several pieces of evidence, raise the possibility of astrocytes playing a crucial role within the FEO system. Firstly, astrocytes are competent circadian oscillators with a temperature-compensated period[32], which can modulate clock gene expression of other cell types such as neurons[24,25,33]. Secondly, in line with the function of astrocytes as metabolic sensors of systemic cues[2], our results in vitro demonstrate that INS and glucose can shift astrocyte rhythms in gene expression. Thirdly, we observed that alongside the impaired INS-induced shift in FAA, *IRcKO* male mice also exhibited a marked reduction in PER2 expression. Lastly, *IRcKO* mice displayed diminished DA release within the striatum, and the administration of a DRD2 agonist induced FAA shifts in the mutants. This suggests a potential role for astrocytic INS signaling in modulating dopaminergic neurotransmission to fine-tune the anticipation of scheduled feeding. Given the complex nature of the FEO system, further investigation involving direct astrocyte-neuron interactions within FEO-relevant brain regions, such as the striatum and hypothalamus, is warranted. These investigations would provide additional support for the contribution of astrocytes to the intricate functioning of the FEO.

We postulate that changes in FAA and body clocks likely require the activation of astrocyte INS signaling after feeding. It might be argued that since INS levels rise postprandially, this could make it difficult to explain the INS effects on FAA during fasting[8]. However, we identified rhythmic transcripts related to INS signaling and dopaminergic signaling in primary astrocytes synchronized with DEX, suggesting that the activation of IR is under the control of the astrocyte´s clock. Additionally, INS stimulates ATP release from astrocytes, which acts on purinergic signaling in DA neurons[31]. Notably, astrocytes release ATP in a circadian manner[88], suggesting that enhanced INS sensitivity at a specific time of the day could drive ATP secretion from this glial cells, influencing neuronal activity. It is worth noting that INS sensitivity and efficient glucose uptake are intrinsically controlled by the clock[2]. Furthermore, proper glucose entry into the brain relies on astrocytic INS signaling[30]. Our investigation also unveils rhythmic fluctuations in glucose transporter expression, including GLUT1,

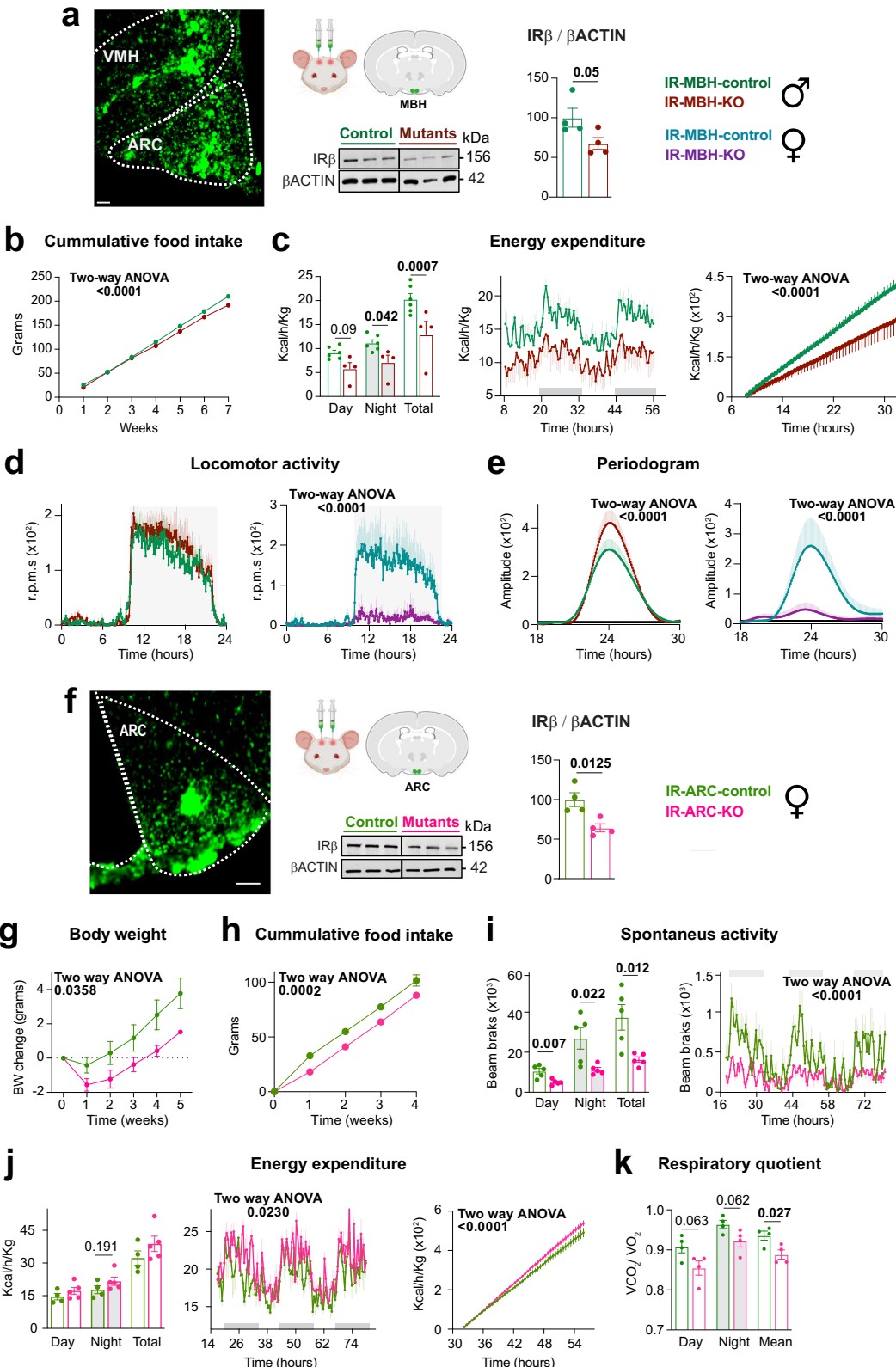

highly expressed by astrocytes[41], in the hypothalamus. Elevated levels of phosphorylated AKT (pAKT) observed at ZT0 align with diurnal brain IR sensitivity regulation, further emphasizing the role of astrocytes in orchestrating this intricate process. The correlation between plasma INS and cerebrospinal fluid (CSF) INS levels[89–92], coupled with observed 24-h brain glucose utilization rhythms[37,38], underscores intricate interactions among astrocytic INS signaling, circadian

rhythms, and metabolic activities. This substantiates our proposition that the observed changes in FAA and circadian rhythms hinge on dynamic activation of astrocytic INS signaling post-feeding.

Although we observed similar IR recombination in the SCN and ARC of both males and females, the reduction in IRβ within the hypothalamus of the mutants was more subtle in females. It plausible that distinct cellular populations within the hypothalamus could show

**Fig. 7 | Astrocytic IR in the MBH regulates circadian behavior and energy balance. a** MBH showing AAV-GFAP-GFP infection in $IR^{fl/fl}$ mice. Scale bar: 100 µm. IRβ expression in the MBH of AAV-GFAP-GFP (IR-MBH-control) or AAV-hGFAP-Cre (IR-MBH-KO) mice ($n = 4$). Two-tailed, unpaired $t$-test. **b** Cumulative food intake in IR-MBH-control ($n = 6$) and IR-MBH-KO ($n = 5$) animals. Two-way ANOVA. **c** Hourly plots show hourly energy expenditure, and bar plots show total energy expenditure in controls ($n = 6$ mice) and mutants ($n = 4$ mice). Two-tailed, unpaired $t$-test and two-way ANOVA. **d** Circadian locomotor activity was monitored in controls and mutants in standard LD cycles for 7–10 days. Activity waveforms are shown for male and female control ($n = 5$ mice) and mutants ($n = 6$ males and $n = 5$ females). Activity counts are expressed as the average amount of activity in 5-min bins. Data are plotted with nighttime hours from 8 to 20 and given in ZT. Two-way ANOVA.

**e** Lomb–Scargle periodograms of male and female control ($n = 5$ mice) and mutant mice ($n = 6$ males and $n = 5$ females). Two-way ANOVA. **f** ARC showing AAV-GFAP-GFP infection in $IR^{fl/fl}$ mice and IRβ expression in the ARC of IR-ARC-control or IR-ARC-KO mice ($n = 4$). Two-tailed, unpaired $t$-test. Scale bar: 100 µm. Body weight change ($n = 4$ mice) **(g)** and cumulative food intake ($n = 5$ mice) **(h)** in IR-ARC-control or IR-ARC-KO females. Two-way ANOVA. Hourly plots show hourly spontaneous activity ($n = 5$) **(i)** or energy expenditure **(j)**, and bar plots show total activity **(i)** or energy expenditure **(j)**, in IR-ARC-control ($n = 4$) or IR-ARC-KO ($n = 5$) females. Two-tailed, unpaired $t$-test or two-way ANOVA. **k** Respiratory quotient of IR-ARC-control or IR-ARC-KO females ($n = 4$). Two-tailed, unpaired $t$-test. Data are represented as mean ± s.e.m. Mouse heads and coronal mouse brains (**a, f**) were created with Biorender.com. Source data are provided as a Source Date file.

sex-specific differences in IRβ expression and/or their responsiveness to INS. Notably, our results claim a sexually dimorphic effect of astrocytic INS signaling on the regulation of light and food entrainment of the timekeeping system. This suggest that circadian locomotor activity is unrelated to the higher INS sensitivity of females[93]. Studies in human and animal models indicate that circadian rhythms and the metabolic consequences of circadian disruption show sex differences[62,66]. Specifically, while male rodents show a more remarkable ability to maintain light entrainment[61], females show enhanced entrainment to phase shifts and are more resistant to genetic and environmental circadian disruption[94]. On the other hand, males develop FAA sooner than females and with a higher amplitude of activity[95]. The underlying cause of this sex differences remains unknown but is unrelated to gonadal steroids[95]. Our study consistently demonstrates that altered locomotor phenotypes in *IRcKO* female mice are independent of ovarian function.

We provide valuable insights into the diverse metabolic and circadian outcomes resulting from the ablation of astrocytic IR in specific hypothalamic nuclei, including the MBH, ARC, or VMH. This differentiation becomes more evident when considering the relatively limited impact on the VMH, thereby underscoring the crucial role of ARC astrocytes in governing energy balance and locomotor activity, particularly in females. Our data, confirming the knockout of IRβ in MBH astrocytes through western blot analysis, aligns with our previous findings, which demonstrated a substantial reduction in IRβ mRNA expression in the MBH of the mutants[30]. Additionally, we previously observed a decrease in the percentage of phosphorylated AKT-positive astrocytes in the MBH of the mutants after peripheral INS injection[30]. This evidence further underscores the significance of astrocytic INS signaling in the MBH and its critical role in modulating circadian entrainment and locomotor activity. Importantly, the reduction in locomotor activity resulting from the deletion of IR in ARC astrocytes, mirroring the phenotypes observed in MBH-IR-KO and global *IRcKO* females, aligns with previous studies highlighting the vital role of ARC-specific leptin-sensitive neurons in regulating daily locomotor rhythms[96–101]. Despite the absence of circadian variation in leptin receptor expression within these neurons[102], the potential involvement of SCN inputs is evident. The established connection between the SCN and ARC is known to contribute to shaping locomotor patterns[103]. Consequently, the deletion of IRs in ARC astrocytes may reconfigure the network of leptin-sensitive neurons, influencing entrainment to light cues and/or the synchronization of ARC neuronal circuits. Additionally, our findings shed light on the ARC as a recognized circumventricular organ (CVO), a feature that presents interesting prospects. The presence of enriched IR expression within astrocytes in CVOs[104] offers a different perspective on understanding their role in regulating circadian rhythms. For instance, astrocytes within CVOs express vasopressin (AVP) receptors[105], potentially establishing a link between astrocytic activities in the ARC, and the modulation of circadian rhythms through the influence of AVP signaling within the SCN. This intricate interplay involving astrocytes, CVOs, and neuronal circuits associated with circadian rhythms, could shed light on potential

processes underlying the integration of systemic signals with central timing mechanisms. In sum, recognizing the implications for health and the adaptability of circadian rhythms in response to environmental changes, it becomes evident that delving deeper into the regulatory circuits that underlie the metabolic responses to circadian disruption, while considering sex-related factors, offers a promising avenue for future exploration.

We observed a significant alteration in energy balance upon the deletion of astrocytic IRs. Therefore, the impaired shift in the FAA in *IRcKO* mice may involve diminished circadian oscillator entrainment or altered metabolic homeostasis. As the connection of metabolism and the circadian clock works in both directions, it is not surprising that animal models with genetic clock defects in astrocytes display metabolic alterations and that clock alterations can be found in metabolically challenged conditions[2]. Despite this fact, identifying that INS signaling in astrocytes has sexually dimorphic roles in regulating circadian rhythms and energy homeostasis is particularly noteworthy. Indeed, our findings provide valuable insights into the development of sex-based therapies to prevent metabolic disorders associated with circadian misalignment based on astrocyte-targeted drugs. Moreover, it will have important repercussions for further understanding the health benefits of time-restricted eating (intermittent fasting) in weight-loss therapy.

## Methods
### Animals and treatments
Sprague Dawley Rats (crl:SD 400, Charles River Laboratories, Barcelona, Spain) were used in all experiments involving rats. Female and male postnatal day 1–3 Sprague Dawley Rats were housed with their dams in a 12 h light–dark cycle in a temperature and humidity-controlled room. Water and food (Teklad-7913, Envigo) were provided ad libitum for the dam. Mice having the sequence of the IR gene flanked by loxP sites ($IR^{f/f}$) (generated by Ronald Kahn, Joslin Diabetes Center) were crossed with the Glast[CreERT2] mouse line[54] and with the Rosa26 ACTB-tdTomato/EGFP (tdTomato/eGFP) reporter mice (Strain #:007676; RRID: IMSR_JAX:007676; Jackson Laboratory) (*IRcKO* mice). Mice were housed with ad libitum access to food and water and kept on a 12-h (8 a.m. to 8 p.m.) light-dark cycle in a temperature- and humidity-controlled room at the animal facility of the University of Santiago de Compostela.

Care of animals was within institutional animal care committee guidelines. All procedures were reviewed and approved by the University of Santiago de Compostela Ethics Committee following the European Union normative for the use of experimental animals (Project ID 15012/2021/011).

Eight to ten weeks old Glast[CreERT2]; IR[f/f] (*IRcKO*) and controls ($IR^{f/f}$) were treated with Tamoxifen (Sigma-Aldrich, T-5648) dissolved in corn oil (Sigma-Aldrich, C-8267)[24,25,29,55]. Animals received 5 mg/day for 2 days by oral gavage[24,25,29,55].

Mice were allowed ad libitum access to water and standard chow (Teklad-7913, Envigo) or HFD (Research Diets Inc D12492, 60% fat, 20% carbohydrate, 20% protein, 5.21 kcal/g) starting 1 week after the

TM treatment. Body weights and food intake were determined weekly.

The DRD2 agonist quinpirole (1 mg/kg body weight, Sigma-Aldrich) was administered daily at ZT12 to the animals by intraperitoneal injection for a duration of 5 days. This method of administration and dosage was selected based on previous studies and its known effectiveness in modulating DRD2 activity in mice[70,106].

The whole-body composition was measured using nuclear magnetic resonance imaging (Whole Body Composition Analyzer; EchoMRI, Houston, TX). Body length, from tip of nose to base of tail was obtained using a digital camera. Skin temperature surrounding BAT was recorded with a B335 compact infrared thermal imaging camera (FLIR) and analyzed with FLIR Tools software (FLIR Systems)[107–110]. Infrared thermography images were taken from the back of the mice to precisely visualize heat production from the BAT. For each image, the BAT surrounding area was delimited, and the average temperature of the skin area was calculated as the average of 3 pictures/mice.

### Indirect calorimetry
Mice were analyzed for energy expenditure, respiratory quotient, and spontaneous locomotor activity using a calorimetric system (Lab-Master; TSE Systems)[29,63,107,109,111]. Mice were acclimated for 24 h to the test chambers and then were monitored for an additional 48–72 h.

### Ovariectomy
Three weeks after TM treatment, control and *IRcKO* mice were bilaterally ovariectomized (OVX) or sham-operated[29,63–65]. Briefly, under ketamine–xylazine anesthesia (50 mg/kg, intraperitoneal), both flanks were shaved, and mice were placed on their left flank. Then, an incision was made in the right flank skin and muscle layers. The ovary was then carefully pulled out. The distal part of the uterine horn was ligated with a surgical suture, and then the ovary was removed after making a dissection between the suture and the ovary. After returning the horn to the abdominal cavity, the wound was closed, suturing the muscle layer was with surgical silk and applying surgical staples to the skin. The same procedure was used to remove the left ovary. Sham surgeries were also performed, in which each ovary was exposed but not tied or dissected. All experimental tests on OVX mice were carried out 4 weeks after surgery to ensure a total washout of ovarian hormones[29,63–65].

### Stereotaxic microinjection of adeno-associated virus (AAV)
To ablate IR specifically in mediobasal astrocytes, we used AAV viral particles (serotype 2/5) expressing GFP or Cre protein under the control of the hGFAP promoter (Vector Biolabs, Philadelphia, USA). We stereotaxically injected AAV-hGFAP-GFP (controls) or AAV–hGFAP-Cre (mutants) particles bilaterally ($2 \times 10^9$ viral genome particles per side) into the MBH, VMH and ARC of *IR*[f/f] littermates by using a 33-gauge needle connected to a 1 ml syringe (Neuro-Syringe, Hamilton). Stereotaxic coordinates were −1.5 mm posterior and −0.4 mm lateral to bregma, and −5.8 mm ventral from the dura for the MBH; −1.5 mm posterior, ±0.2 mm lateral to bregma and −6 mm dorsoventral for the ARC; −1.7 mm posterior and −0.5 mm lateral to bregma, and −5.5 mm ventral from the dura for the VMH. Before needle retraction, a 10-min time-lapse was allowed to prevent backflow through the needle track before the syringe was withdrawn. After the procedure, the skin incision was closed with surgical sutures (Henry Schein), and mice were placed in a heated cage until they recovered from anesthesia. Experiments were conducted at least 3 weeks after injections to ensure AAV expression. Surgeries were performed using a mixture of ketamine and xylazine (15 mg/kg and 3 mg/kg). Meloxicam (5 mg/kg) and buprenorfin (0.05 mg/kg) were used as analgesic agents.

### Circadian locomotor activity
Four to five months old, *IRcKO* and control mice were single-housed in cages equipped with running wheels[24,25,29] (ENV-044; Med Associates Inc). To avoid potential interference between male and female mice, we employed separate ventilated cages placed in isolated racks, each with its own ventilation system. This arrangement prevented direct contact and olfactory interaction between sexes, thereby eliminating the possibility of male presence affecting female circadian entrainment. For the light-dark experiments, we maintained uniform lighting conditions for both sexes by housing them in the same room and arranging them similarly in the racks, ensuring consistent lighting conditions throughout the study. Mice were acclimated to the running wheels for 3 days in standard light-dark cycles (12 h:12 h) before the experiment commenced, which was conducted under the same conditions for 7–10 days. Subsequently, the mice were placed in isolated black cages (Tecniplast), for 3–4 weeks under constant darkness, followed by 7–10 days of exposure to light-dark cycles (12 h:12 h[24,25,29]. Running wheel activity was recorded in 5-min bins by Wheel Manager software (SOF-860; Med Associates Inc). The data obtained were analyzed with Actogram J[24,25,29].

The daily locomotor activity patterns were extracted from a minimum of 7–10 days of wheel-running data per mouse, ensuring a comprehensive representation of the overall daily pattern. This approach effectively accounted for potential variations due to the estrous cycle in female mice. The activity profiles during light-dark (LD) and re-entrainment light-dark (rLD) conditions were generated by plotting data from nighttime hours (8–20) using *zeitgeber* time (ZT) as the reference. In contrast, activity profiles during constant darkness (DD) were adjusted to accommodate individual variations in period lengths. Specifically, the abscissa units were represented in circadian time (CT). The mean activity was computed by averaging the activity data within 5-min intervals across each individual's circadian cycle.

### Food restriction experiments
Control and *IRcKO* mice, after 6 weeks of TM treatment, were single-housed in cages equipped with running wheels[24,25,29] (ENV-044; Med Associates Inc). Before RF schedules, locomotor activity was monitored in LD for 7 days under *ad libitum* feeding conditions. Body weight and food intake were determined at ZT6, after which mice were exposed to a restricted feeding schedule for 10 days. Under this schedule, access to standard lab chow (ENVIGO Company) was restricted from ZT4 to ZT10 for 6 days and from ZT4 to ZT8 for the next 4 days. To monitor food consumption, food pellets for each mouse were weighed daily before (ZT4) and after food availability (ZT10 or ZT8). Running wheel activity was recorded in 5-min bins by Wheel Manager software (SOF-860; Med Associates Inc). The data obtained were analyzed with Actogram J[24,25,29].

### Primary hypothalamic astrocyte cultures
Primary monolayer cultures of astrocytes were established from the hypothalamus of neonatal (P1–P3) Sprague Dawley rats. Neonatal rat pups were decapitated using sharp scissors. The astrocyte cultures were maintained at 37 °C in a humidified atmosphere of 5% $CO_2$ for 1 week[25,33,112–114]. After that, cells were trypsinized and subcultured for the experiments[25,33]. These astrocyte-enriched cultures contained >96% astrocytes as indicated by immunofluorescence, with a monoclonal antibody anti-GFAP (Clone 6F2, Dako; dilution 1:1000)[25,33,112–114]. Astrocytes were synchronized with 100 nM of Dexamethasone or 600 nM of INS[22,25,33] for 1 h. The hormones were washed out and cells were harvested at different time points for subsequent analyses[25,33].

### RNA isolation and quantitative real-time RT-PCR
Cells were harvested at the appropriate time points. For each time point, we prepared samples for the assay in duplicate or triplicate[25,33]. Total RNA was extracted using TRIzol reagent following the manufacturer's instructions. RNA was further cleaned using a RNeasy Mini Kit. Complementary DNA was obtained by retrotranscribing 0.5 μg of total mRNA using the ImProm-II Reverse Transcription System

following the manufacturer's instructions. Real-time reverse transcriptase–PCR was done using the ABI PRISM.7900 (Applied Biosystems). For a 15 µl reaction, 9 ng of cDNA template was mixed with the primers to a final concentration of 200 nM and mixed with 7.5 µl of 2 × QuantiFast SYBR Green PCR Master Mix. The reactions were done in duplicates using the following conditions: 5 min at 95 °C followed by 40 cycles of 10 s at 95 °C, 30 s at 60 °C and 1 min at 70 °C. The sequence of the primers used is provided in Supplementary Data 5. *Gapdh* transcripts were used as reference controls[25,33].

## Western blotting

Brown adipose tissue, hypothalamus, MBH and ARC were homogenized in lysis buffer (consisting of a mix of 0.05 M Tris-HCl, 0.01 M EGTA, 0.001 M EDTA, 0.016 M Triton X-100, 0.001 M sodium orthovanadate, 0.05 M sodium fluoride, 0.01 M sodium pyrophosphate and 0.25 M sucrose, made up with distilled water and adjusted to 7.5 pH; all of them from Sigma; St. Louis, MO, USA) and freshly added protease inhibitor cocktail tablets (Roche Diagnostics; Indianapolis, IN, USA). The protein concentration was determined by the Bradford Method (Protein assay dye concentrate, Bio-Rad Laboratories; Hercules, CA, USA), and the total protein content of the tissues was calculated. The protein lysates were subjected to SDS-PAGE and electrotransferred onto a nitrocellulose paper[112,114] (Protran, Schleicher and, Schuell, Dassel, Germany). Membranes were then probed with antibodies against PER2 (Invitrogen, PA5-89045) (dilution 1:1000); CRY1 (Invitrogen, PA5-89349) (dilution 1:1000); IRβ (CT-3) (Santa Cruz Biotechnology, sc-57342) (dilution 1:500); UCP1 (Abcam, ab10983) (dilution 1:1000); phospho-mTOR (Cell Signaling, 2971-S) (dilution 1:1000); mTOR (Cell Signaling, 2972-S) (dilution 1:1000); phospho-AKT (Cell Signaling, 9271) (dilution 1:1000); AKT (Cell Signaling, 9272) (dilution 1:1000); GLUT1 (Invitrogen, PA1-46152) (dilution 1:1000)[25,29,63,115,116]. Horseradish peroxidase-coupled secondary antibodies were purchased from Santa Cruz Biotechnology and used at 1:5000 dilution (sc-2357 and sc-525408).

Membranes were then extensively washed and re-probed with antibodies against GADPH (Santa Cruz Biotechnology, sc-25778) (dilution 1:5000); β-actin (Sigma-Aldrich, A5316) (dilution 1:5000) or α-tubulin (Sigma-Aldrich, AB_330337) (dilution 1:5000), as the loading controls. Immunoreactive bands were detected with a western light chemiluminescence detection system (ECL, GE Healthcare BioSciences AB). Autoradiographic films (Fujifilm, Tokyo, Japan) were scanned, and the band signal was quantified by densitometry using ImageJ 1.33 software (NIH)[25,63,107,109,110,112–116].

Values were expressed in relation to GAPDH (primary astrocytes), β-actin (hypothalamus), and α-tubulin (BAT). Representative images for all proteins are shown; in the case of the loading controls, a representative gel is displayed, although each protein was corrected by its internal control (GAPDH, β-actin, or α-tubulin). In all the figures showing images of gels, all the bands for each picture are from the same gel, although they may be spliced for clarity.

## Immunofluorescence

Mice were anesthetized with ketamine/xylazine (150 mg/kg, 10 mg/kg) and perfused transcardially with ice-cold PBS followed by ice-cold 4% PFA[24,25,29,55]. The brains were post-fixed overnight in 4% PFA in PBS, and 30 µm thick slices were prepared using a cryostat (Leica). The slices were mounted with Prolong Gold and imaged using an inverted laser scanning confocal microscope (Leica TCS SP5) with a 20x or 40x objective (Leica Microsystems). Quantification and analysis were performed using ImageJ software (Wayne Rasband, NIH, USA) by outlining the hypothalamus from the Dapi-stained image and measuring the relative intensity of the fluorescence. When multiple sections were analyzed from each animal, the mean of the measurements from consecutive sections was used for that individual.

## Immunohistochemistry

BAT samples were fixed in 10% formalin buffer for 24 h and subsequently dehydrated and embedded in paraffin following a standard procedure. Sections of 3 µm thickness were prepared using a microtome and stained using a standard Hematoxylin/Eosin Alcoholic procedure (BioOptica) as per the manufacturer's instructions. UCP-1 immunohistochemistry was performed using a rabbit anti-UCP-1 antibody (dilution 1:2000; Abcam). UCP-1-positive cells were quantified using ImageJ analysis software[72,107,109,117].

## Glucose tolerance test

Glucose tolerance test (GTT) was performed after an intraperitoneal injection of $2 \, g/kg^{-1}$ D-glucose (G8270, Sigma-Aldrich), as shown[24,29,110,117,118].

## Determination of metabolic markers

Cholesterol (no. 1001093, Spinreact) and triglycerides (no. 1001314, Spinreact) circulating levels were measured by spectrophotometry in a Multiskan GO spectrophotometer (Invitrogen-Thermo Fisher). Leptin and IGF1 serum levels were measured using ELISA with reagent kits and protocols provided by Merck-Millipore (leptin, EZML-82K) and Thermo Fisher Scientific (IGF1, EMIGF1).

## High performance liquid chromatography (HPLC)

WT and *IRcKO* mice were sacrificed by rapid decapitation to preserve the unstable nature of monoamines including dopamine, noradrenaline, and serotonin. The hypothalamus and striatum were rapidly dissected on an ice-cold plate, immediately frozen on dry ice, and stored at −80 °C until analysis[72,73]. Then tissue was homogenized and then centrifuged ($14,000 \times g$; 10 min at 4 °C). The remaining supernatant fraction was filtered and injected (20 µl/injection) into the High-Performance Liquid Chromatography (HPLC) system (Shimadzu LC Prominence; Shimadzu Corporation; Kyoto, Japan)[71,73,74]. Dopamine and its metabolites 3,4-dihydroxyphenylacetic acid (DOPAC) and homovanillic acid (HVA), serotonin (5-HT) and its metabolite 5-hydroxyindolacetic acid (5-HIAA), and noradrenaline (NA) were separated on a reverse phase analytical column (Waters Symmetry300C18; 150 × 3.9 mm, 5 µm particle size; Waters). The mobile phase (70 mM $KH_2PO_4$, 1 mM octanesulfonic acid, 1 mM EDTA, 10% MeOH, pH 4) was delivered at a rate of 1 ml/min. Detection was performed with a coulometric electrochemical detector (ESA Coulochem III). The first and second electrode of the analytical cell were set at +50 mV and +350 mV, respectively; the guard cell was set at −100 mV. Data were acquired and processed with Shimadzu LCsolution software. Results were expressed in nanogram per milligram of tissue (wet weight). The DOPAC/DA and DOPAC + HVA/DA ratios were calculated for each animal as an index of the DA turnover.

## In silico analysis

To examine genome-wide circadian gene expression in synchronized astrocytes, cells exposed to 0.25, 6, 12, 18, and 24 h of DEX treatment were used for the analysis. Expression data were subjected to BIO_CYC cosine analysis[119,120] to identify transcripts expressed within a ~24 h period. We assumed that transcripts whose expression levels have a circadian component would show one complete oscillation every ~24 h. Transcripts targeted by probes identified as overt circadian were defined as having a prevalent circadian expression in the associated test condition. We used ns to keep the false-discovery rate (FDR) to a maximum of 5%[121]. Illumina Probe IDs were submitted to DAVID Functional Annotation Clustering[122,123] or manually analyzed using the National Center for Biotechnology Information (NCBI) gene database to update annotations and identify single genes associated with multiple regulated probes (http://www.ncbi.nlm.nih.gov/gene); for probes that targeted the same gene, the probe containing the highest magnitude of changes was used for downstream analyses. Rhythmic genes

were plotted by peak phase and were then separated into phase clusters. Transcript median profiles (median across all paired samples, per sampling time point, of z-scored time series) were entered in the clustering analysis. The number of clusters was established using the Bayesian index criterion[50]. We conducted gene ontology (GO) analyses with the DAVID Bioinformatics Resources[122,123] to classify the rhythmic transcripts into Kyoto Encyclopedia of Genes and Genomes (KEGG) pathways. To identify more specific GO terms (versus broad), we performed analyses using default settings for terms of the GO biological processes (BP_FAT).

## Statistical analysis

Data are presented as mean ± SEM and were analyzed and graphed using Prism 9 (GraphPad, San Jose, CA, USA). Statistical parameters, including the exact value of *n* and precision measures (mean ± s.e.m.) and statistical significance, are reported in the figures and figure legends. Statistical significance was assessed using several methods. Statistical significance was evaluated using several methods. Two-tailed paired Student's *t* tests were employed for comparing paired data points. One-way analysis of variance (ANOVA) was used, followed by Tukey–Kramer post hoc tests, when comparing multiple groups. Additionally, two-way repeated measures ANOVA was utilized, followed by Bonferroni post hoc tests, for cases involving repeated measurements. Data were checked for normality and equal variances between groups.

The statistical significance of the rhythmic expression was determined by Cosinor analysis[24,25,29,33,55]. Samples or animals were excluded from the data analysis with pre-established criteria if they deviated more than two s.d. from the group mean. In the food-restricted experiments, activity data were quantified by calculating pre-feeding activity duration (in hours from the pre-feeding activity phase to mealtime). FAA ratio was calculated by the fold change of mean locomotor activity in each mouse during the 4 h period before feeding compared with the rest of the day. The nocturnal activity ratio was calculated by the fold change of mean locomotor activity in each mouse during the night before providing compared with the rest of the day.

## Reporting summary

Further information on research design is available in the Nature Portfolio Reporting Summary linked to this article.

## Data availability

All data generated in this study are provided in the Source Data file. The microarray data used in this study are available in the NCBI Gene Expression Omnibus database under accession code GSE39272. Source data are provided with this paper.

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

## Acknowledgements

We thank Dr. M. Götz (Physiological Genomics, Biomedical Center, Ludwig-Maximilians-University Munich, Germany) for kindly providing the GLAST^CreERT2 mouse line. We thank CiMUS technical staff for their excellent support. We also thank the Animal Facility of USC (CEBEGA), for assistance in animal experiments. The research leading to these results has received funding from: Xunta de Galicia, Consellería de Cultura, Educación e Ordenación Universitaria (O.B.M.: ED431F 2020/009 and ED431C 2023/28); Agencia Estatal de Investigación (O.B.M.: PID2019-109556RB-I00; PID2022-138436OB-I00); Ministerio de Ciencia e Innovación co-funded by the FEDER Program of EU (M.L.: PID2021-128145NB-I00); O.B.M. is supported with a Ramón y Cajal award (RYC2018-026293-I) from the Ministerio de Ciencia, e Innovación of Spain. C.G.C. is supported by funding from the European Research Council ERC (C.G.C.: STG grant AstroNeuroCrosstalk # 757393), the German Research Foundation DFG under Germany's Excellence Strat-egy within the framework of the Munich Cluster for Systems Neurology (EXC 2145 SyNergy—ID 390857198), and the Helmholtz Association—Initiative and Networking Fund. CiMUS is supported by the Xunta de Galicia (2016-2019, ED431G/05). CIBER de Fisiopatología de la Obesidad y Nutrición is an initiative of ISCIII. M.L.M. and M.S.L. are supported from the Ministerio de Ciencia, Innovación y Universidades of Spain (PRE2020-093614 and FPU2018/00647). The funders had no role in study design, data collection and analysis, decision to publish, or pre-paration of the manuscript. We would like to acknowledge that images were adapted from "Mouse brain (coronal cut)", "Mouse head" and "Syringe" by BioRender.com (2023). Retrieved from https://app.biorender.com/biorender-templates.

## Author contributions

A.G.V., M.L.M., M.S.L., N.O. and M.G.D. performed the in vivo experi-ments and tissue analysis. A.G.V. performed the in vitro experiments. P.G.G. performed the determination of DA and metabolites by HPLC. O.B.M., M.L., J.L.L.G. and C.G.C. conceived, designed the experiments, and coordinated the project. O.B.M. and M.L. secured funding and supervised the work. O.B.M. wrote the manuscript and made the figures. O.B.M. is the lead contact. All authors analyzed, interpreted, discussed the data, and revised and edited the manuscript.

## Competing interests

The authors declare no competing interests.
