## [Peer Review File · Nature Communications]

REVIEWER COMMENTS

Reviewer #1 (Remarks to the Author):

The article entitled "Astrocytic insulin receptor controls circadian behavior via dopamine signaling in a sexually dimorphic fashion" submitted by Lopez and Barca-Mayo explores the role of astrocytic insulin receptor on light- and food-dependent entrainment of circadian rhythms, interestingly assessing a sex-dimorphic effect.

The data reported is potentially sound, however, this reviewer opinion's is that some critical methodological aspects detailed below should be addressed for a better interpretation of the data and before considering the work suitable for publication.

On the significance to the field and originality of the work

Undoubtedly, the role of astrocytes as timekeepers has just started to be explored in our field and it is extremely relevant, to understand how circadian physiology is regulated at central level. Uncovering mechanisms controlling light and food entrainment, considering sex differences, are critical to design strategies that reduce the metabolic and behavioral effects of circadian disruption.

The urgency is also unquestionable since our current life style favors circadian disruption, by demanding the exposure to artificial light all night, work in night-shifts, travelling across time zones, and spending several hours in front of screens at night.

On technical/methodological aspects and data supporting author's conclusions

1) Systemic insulin receptor sensitivity is diurnally regulated, peaking upon waking. Is this also true at the level of the hypothalamus/hypothalamic astrocytes? Demonstrating whether this is the case might be relevant for data interpretation.

2) Fig 1: Is the insulin concentration used in cell culture within a physiological or pharmacological range? How was it selected?

3) If the intention is to compare the effect of clock gene expression between DEX and INS and argue about the in vivo situation (lines 124-126), the data from Fig 1 might not be sufficient because in vivo there is a close interaction between GR and IR signaling pathways that are both clock-phase dependent.

4) The effect of INS and INS+glucose could be better described for Cry1.

5) When validating the recombination efficiency in IRcKO mice, did authors check whether there is any sex difference on the recombination rate? i.e. are the western blot results showed in Fig 2A also true in females?

7) The main concern of this reviewer is related to the interpretation of the sexual dimorphic effect. Several methodological aspects need to be clarified here.

- Was female estrous cycle checked or synchronized? It is relevant to explain how the housing of males and females was done during the running-wheel experiment, were females exposed to male's odor? It has been shown that female's light entrainment changes over the estrous cycle and that the cycle can be influenced by the presence of male's odor. Knowing these methodological details are critical for data interpretation specially because it is not clear from the text and from the figure legend whether the daily pattern of locomotor activity was calculated as an average of one or several days on each light condition.

- It is also important to know whether the light intensity to which males and females were exposed to during LD experiments was the comparable. One could see from the representative actograms that the offset of activity in males and females is different but without this methodological detail it is difficult to attribute this effect to a sex dimorphism.

8) For the data shown in Fig 3, authors should indicate the way in which they administered Insulin in the FFA shift experiment. It is also important to explain the reasoning behind choosing ZT12 as the time for INS injection. This is relevant to understand whether the effect of the acute manipulation of INS levels is FAA phase-dependent.

9) The weight and body composition changes shown in Fig 4 could have a grow restriction component? Was this possibility evaluated?

10) In Fig 5 authors show data from an alternative genetic manipulation of IR in astrocytes using a viral approach to induce the knock-down, now more specifically in the VMH. In order to interpret the results properly a validation of the AAV approach is necessary.

11) Again, in Fig 5 it is not clear from the text and from the figure legend whether the daily pattern of locomotor activity was calculated as an average of one or several days on each light condition.

12) Authors should also comment on the very reduced locomotor activity observed in females after the viral manipulation (Fig 5C).

13) Please, indicate food access period in panel E (RF 6h and RF 4h).

14) The involvement of dopamine levels explaining how the deficit of IR in astrocytes alters

INS-induced entrainment might be stronger if authors really confirm that dopamine levels are reduced in their IRcKO. Moreover, authors should indicate the way in which they administered DRD2 agonist.

On interpretation and conclusions

Several critical methodological aspects have to be clarified/improved by the authors to better assess whether the data support the conclusions.

Minor concerns

Abstract:

Page 2 line 32, abbreviation for INS has to be explained the first time is mentioned.

Introduction:

Page 3 line 57: One can argue that shift work not only induces mistimed feeding but also mistimed light exposure and both might disrupt circadian physiology.

Page 3 lines 69-71: The fact that 20% of oscillating proteins do not show signs of rhythmicity at mRNA level suggests that rhythmic regulation of transcription is just one aspect but post-transcriptional, translational, post-translational protein stability, subcellular localization are also regulated rhythmically by the molecular clock.

Page 3 line 71: the word clock seems to be missing after molecular

Page 3 line 75: circadian clock has to be deleted

Results:

Page 4 line 90: *in vitro* goes in italics

Page 5 line 120: "astrocyte clock" has to be changed by "astrocyte's clock"

Page 5 line 126: "onset of increase feeding" has to be changed by "onset of activity"

Page 5 line 130: *in vivo* goes in italics

Page 6 line 165: IRcKO female and male?

Page 6 line 169: Zeitgeber time goes in italics

Page 6 line 174: The sentence has to be re-written, as a suggestion: To uncover the contribution of astrocyte's INS signaling on the regulation of circadian locomotor activity

Page 7 line 206: did authors mean male and female mutants here?

Page 8 line 219-220: should it read "...this effect was absent in IRcKO female mice"?

Page 8 line 232: locomotor rhythms sounds more accurate than behavioral rhythms here.

Page 10 line 276: The sentence has to be re-written, as a suggestion:the role of VMH

astrocytes's INS signaling in sustaining...

Page 11 line 315: The word "astrocytes" is repeated 3 times in the same sentence

Page 11 line 318: the word "and" y repeated twice

Discussion:

Page 13 line 372: *in vitro* goes in italics

Methods:

Page 14 lines 414: *ad libitum* goes in italics

Reviewer #2 (Remarks to the Author):

In the present manuscript Gonzalez Vila et al. perform a solid and deductive series of experiments to demonstrate that a limiting factor for the control of circadian behavior is the role of the insulin receptor in astrocytes. This regulation occurs in a sexually dimorphic manner and is mediated by dopamine signaling. Identifying this critical role in astrocytes is fundamental to further understand components of one of the most complex and most essential processes for survival, food intake. Given the exponential rise in obesity and pathologies associated to energy balance dysregulations, it is critical to further understand in a holistic manner and including all the pieces of the bigger picture machinery the feeding process. This manuscript brings us a step closer in understanding how our body responds to temporal shifts in feeding regulation which frequently happen in the human population.

The manuscript is solid, data is adequately analyzed following appropriate statistic tests and is of interest to a broad readership. Taken together the strength of these new sets of data, together with the scope of the journal I believe this manuscript is a fit for Nature Communications.

However, a set of suggestions listed below could elevate the scope and strengthen this manuscript. Most of those points could be answered by expanding the present discussion.

Comments:

Authors demonstrate that insulin plays a fundamental role in regulating circadian rhythmicity in astrocytes. One conceptual question that can be raised is: does the concentration of insulin in the brain oscillate in a daily manner? Is there a link between

obesity (or associated pathologies) and the leakage of the blood brain barrier, consequently affecting the concentration of insulin in the brain? Is astrocyte insulin signaling impaired upon obesity due to insulin resistance?

Diet induced obesity attenuates FAA in mice both in HFD and OB/OB mice. Given that obese subjects are both leptin resistant and insulin resistant, I was wondering whether authors have explored the possibility of leptin signaling being involved in the fundamental role of astrocytes in circadian behavior. For instance, do IRcKO males and females change their entrainment to feeding time when chronically fed a HFD? Is the phenotype still sexually dimorphic?

The reviewer wants to elevate the fact that authors use ANCOVA and point to the regression curves when analyzing energy expenditure. It is an important point that should be seen in all studies doing indirect calorimetry.

Authors elegantly show that IR in the VMH astrocytes (including VMN and ARC) is important for circadian behavior. Could authors narrow down between the VMN and the ARC which is the major contributing cluster. Would the close proximity to a circumventricular organ be a key factor allowing broader sensing due to capillary fenestration and better sensing. Building up on this question, as a discussion point authors could consider suggesting a potential role of astrocytic IR in circumventricular organs being an important part of circadian rhythmicity. Other circumventricular organs such as AP or SFO could be evaluated/discussed.

Given the important role for dopamine signaling for circadian behavior. Do disorders or alterations in dopamine production impair circadian behavior?

Given the sexually dimorphic phenotype, can authors expand on the origin of such dimorphism. Are there sexual hormonal regulation associated to mechanism identified?

We recommend a thorough Spell-check identified before resubmission. Some examples identified in the text.

Line 33: Meaning of INS in the abstract should be detailed

Line 143: When they say figure C, I guess authors refer to figure 1C right.

Line 551: Posthoc instead of postdoc

Reviewer #3 (Remarks to the Author):

Recent work indicates that insulin and dopamine signaling is involved in feeding entrainment of daily rhythms. Here the authors test the role of insulin signaling in astrocytes and whether this is necessary for entrainment of daily rhythms by light and feeding. They find that insulin is sufficient to reset the astrocyte clock, with effects that differ from those induced by glucocorticoids. They also find that eliminating IR in astrocytes modulates activity levels and circadian period in females. Eliminating IR in astrocytes in adulthood also increases FAA acquisition in each sex, but at different phases of food entrainment. Changes in metabolism were also detected after deletion of IR in astrocytes, and these effects were influenced by sex. VMH-specific IR astrocyte deletion altered activity levels and metabolism. DrD2 agonists modulated activity levels. Overall, this work suggests that IR signaling in astrocytes modulates activity levels and metabolism.

Major

1. The individual experiments are technically sound, but the results are not noteworthy. This work will make a larger impact if the causal connections between findings are more clearly articulated.
2. In several places, the authors make inferences that are not clearly supported by evidence. For instance, effects of IR signaling in astrocytes may impact locomotor levels and metabolism by acting through non-SCN circuits, rather than reflecting changes in the SCN neural network (as concluded on lines 188-189, 291-292, 305-306, 310-311, 327). Alternative interpretations are not considered or discussed by the authors. In addition, the authors state that “Our results first suggest that astrocytes are part of the FEO”. The current data deserve a more nuanced interpretation and this claim requires a higher bar of proof than provided by the current work.
3. The DRD2 agonists do not clearly rescue circadian behavior in IRcKO mice. Instead, this

drug had effects in both control and cKO mice.

Minor

1. Viral-mediated IR knockdown in VMH was not verified or quantified.
2. It is not clear if the authors control for familiar wise comparisons (e.g., paired t-test were performed in Figure 1D, which is not appropriate). The use of ANOVA for r.p.m.s and amplitude data in Figures 2, 5, and 6 also seems odd since these data are typically binned and/or summed.
3. It is unclear if activity profiles under DD were corrected for period length (Figure 2B).
4. Re-entrainment after an extended DD free-run is difficult to quantify and interpret given individual differences in period.
5. The authors should use "sex" rather than "gender" when referring to rodents.
6. The authors in several places refer to something as "solid" (e.g., line 339), but it would be preferable for the authors to describe the finding or difference clearly.

REVIEWER#1

Overall comment: The article entitled "Astrocytic insulin receptor controls circadian behavior via dopamine signaling in a sexually dimorphic fashion" submitted by Lopez and Barca-Mayo explores the role of astrocytic insulin receptor on light- and food-dependent entrainment of circadian rhythms, interestingly assessing a sex-dimorphic effect. The data reported is potentially sound, however, this reviewer opinion's is that some critical methodological aspects detailed below should be addressed for a better interpretation of the data and before considering the work suitable for publication.

Undoubtedly, the role of astrocytes as timekeepers has just started to be explored in our field and it is extremely relevant, to understand how circadian physiology is regulated at central level. Uncovering mechanisms controlling light and food entrainment, considering sex differences, are critical to design strategies that reduce the metabolic and behavioral effects of circadian disruption.

The urgency is also unquestionable since our current lifestyle favors circadian disruption, by demanding the exposure to artificial light all night, work in night-shifts, travelling across time zones, and spending several hours in front of screens at night.

Several critical methodological aspects have to be clarified/improved by the authors to better assess whether the data support the conclusions.

Response: We sincerely express our gratitude to Reviewer#1 for dedicating time to thoroughly review our article. The constructive feedback and valuable suggestions provided have been instrumental in enhancing the interpretation of our data and strengthening the overall quality of the manuscript. We addressed the Reviewer's concerns by performing new experiments and discussing or revising the original version of the manuscript. We believe the revised version now presents a clearer and more robust description and interpretation of our research. Below is the point-by-point response to the Reviewer.

MAJOR CONCERNS

Reviewer's major concern 1: Systemic insulin receptor sensitivity is diurnally regulated, peaking upon waking. Is this also true at the level of the hypothalamus/hypothalamic astrocytes? Demonstrating whether this is the case might be relevant for data interpretation.

Response: We genuinely appreciate the reviewer's insightful concern regarding the diurnal regulation of insulin receptor (IR) sensitivity within the hypothalamus, particularly in astrocytes. Addressing this query is indeed essential for contextualizing our findings. The intricacies of the brain's architecture and the challenges posed by the blood-brain barrier make direct measurements of IR sensitivity in specific brain regions or cell types a complex endeavor. Nevertheless, existing evidence underscores the diurnal orchestration of glucose utilization within the brain.

Compelling positron emission tomography (PET) studies consistently highlight a 24-hour rhythm in glucose utilization, demonstrating heightened activity during the dark phase^{1,2}. Intriguingly, this pattern transcends mammalian species and applies across various light-dark and constant darkness conditions^{1,3,4}, with the exception of the SCN^{1,3,4}. The SCN's unique glucose utilization pattern during the light phase has been linked to vasopressin release, influencing glucose transporter 1 (GLUT1) expression in the arcuate nucleus to sustain day-night fluctuations in glycemia⁵. Given the significant expression of GLUT1 in astrocytes⁶, this mechanism might extend to INS sensitivity and glucose utilization in hypothalamic astrocytes.

To address this, we employed quantitative RT-PCR to evaluate the diurnal expression of IR and glucose transporter (*Gluts*) transcripts in the hypothalamus. The results revealed rhythmic oscillations, peaking between ZT12 and ZT18 (**Supplementary Figure 1E**). Additionally, heightened INS signaling at ZT0, evident from elevated pAKT levels, aligns with the diurnal regulation of hypothalamic IR sensitivity (**Supplementary Figure 1F**). Notably, our transcriptome analysis of synchronized astrocytes revealed the enrichment of INS signaling transcripts within a specific cluster, further supporting the diurnal modulation of IR sensitivity in the hypothalamus, potentially involving astrocytic contributions (**Supplementary Figure 1D and Supplementary Table 1**).

We acknowledge the reviewer's assertion that understanding the diurnal variation of IR sensitivity in hypothalamic astrocytes bears paramount significance due to the brain's reliance on glucose for energy and its susceptibility to glucose-related disruptions. While the direct measurement of IR sensitivity in these astrocytes poses challenges, innovative imaging techniques like functional magnetic resonance imaging (fMRI) or PET hold promise for future investigations. The outcomes of such studies may unveil potential therapeutic avenues. In response to the reviewer's feedback, we have thoughtfully included a discussion on the diurnal regulation of IR sensitivity in hypothalamic astrocytes. We genuinely value the reviewer's thoughtful insights, and we are committed to integrating these considerations into our forthcoming research endeavors.

Reviewer's major concern 2: Fig 1: Is the insulin concentration used in cell culture within a physiological or pharmacological range? How was it selected?

Response: We greatly value the reviewer's critical assessment of our study's methodology, particularly concerning the INS concentration employed in our cell culture experiments. We acknowledge the importance of meticulously choosing appropriate INS doses, recognizing that *in vitro* concentrations may not perfectly mirror physiological or clinical scenarios due to inherent variability in individual INS sensitivity and glucose levels.

The range of INS concentrations utilized in *in vitro* studies spans from picomolar to micromolar levels. Lower concentrations are commonly employed to explore initial signaling events, while higher concentrations are chosen to trigger downstream effects or induce INS resistance. In our study, the selection of the INS concentration was guided by a thorough review of well-established literature. Our guiding reference was the work of Crosby et al.⁷, whose findings indicated that even at a modest concentration of 1 nM, INS significantly influenced cellular rhythms. This concentration effectively modulated the circadian phase, period, and amplitude of PER2 in diverse cell types, including fibroblasts, primary cortical neurons, organotypic liver and kidney slices, and intestinal organoids derived from PER2::LUC mice⁷. For our *in vivo* experiments, the selected INS concentration (2.25 IU/kg)⁷ falls well within the established normal range typically employed in mouse studies⁸⁻¹⁰.

We genuinely appreciate the raised concern, as it underscores the importance of considering contextual factors like INS sensitivity and glucose levels when interpreting our study's outcomes. We are confident that this explanation addresses your concerns and provides a clear understanding of our methodological rationale.

Reviewer's major concern 3: If the intention is to compare the effect of clock gene expression between DEX and INS and argue about the *in vivo* situation (lines 124-126), the data from Fig 1 might not be sufficient because *in vivo* there is a close interaction between GR and IR signaling pathways that are both clock-phase dependent.

Response: We appreciate your thoughtful concern about our comparison of clock gene expression between DEX and INS treatments and its relevance to the *in vivo* context. Recognizing the intricate interactions between the GR and IR signaling pathways, both influenced by the circadian clock and potentially yielding phase-dependent effects on clock gene expression, is essential. While our *in vitro* data indicate a delay in clock gene expression with DEX compared to INS stimulation in astrocytes, we recognize that directly extrapolating these results to the *in vivo* setting is complex. To address this concern, we have emphasized the multifaceted dynamics of the hypothalamic-pituitary-adrenal axis and the temporal interplay between GR and IR signaling pathways in the revised manuscript (page 5, lines 128-132). This highlights the need for a nuanced consideration of *in vivo* complexities. We greatly value your feedback as it has contributed to enhancing the interpretation and significance of the findings from our study.

Reviewer's major concern 4: The effect of INS and INS+glucose could be better described for *Cry1*.

Response: We are grateful for highlighting the need for a more detailed description of the effects of INS and INS + glucose on *Cry1* expression. To address this, we have expanded our explanation to provide a clearer understanding of this crucial aspect of our study. Moreover, we have conducted a novel experiment involving primary hypothalamic astrocytes synchronized with INS under low and high glucose levels, to examine their impact on CRY1 protein levels (Figure 1C). Our findings reveal that astrocytes synchronized with INS under these conditions exhibit rhythmic oscillations in CRY1 expression, aligning well with the observed patterns in *Cry1* mRNA levels (Figures 1A and 1C). Additionally, we explored the effects of shorter durations of DEX and INS treatments in low glucose media (1, 2, 3, and 4 hours) on CRY1 protein levels in primary hypothalamic astrocytes. Notably, INS treatment resulted in a decline in CRY1 protein levels after 2 hours, while the presence of glucose led to elevated CRY1 protein levels at 1 and 2 hours (Figure 1C). These intricate findings underscore the complexity of CRY1 regulation by INS and glucose, emphasizing the dynamic interactions within the molecular clock of astrocytes. Notably, these results are consistent with prior research, indicating the influence of INS and glucose on *Cry1* mRNA and protein levels, across diverse tissues and cell types^{11,12}.

We believe that these revisions effectively address your concern and provide a more comprehensive and coherent elucidation of how INS and glucose influence *Cry1* expression.

Reviewer's major concern 5: When validating the recombination efficiency in IRcKO mice, did authors check whether there is any sex difference on the recombination rate? i.e. are the western blot results showed in Fig 2A also true in females?

Response: We appreciate the reviewer's insights into potential sex-specific variations in recombination efficiency and INS signaling within our study. In response, we extended our investigation to include female *IRcKO* mice and examined hypothalamic IR β and phospho-mTOR expression at ZTO using western blotting. In the mutants, observed a reduction in IR β levels that, while not achieving statistical significance, exhibited a decrease in phospho-mTOR levels. Moreover, there was a decrease in PER2 expression and a slightly diminished CRY1 expression (Supplementary Fig. 2B). It's conceivable that this might stem from sex-specific fluctuations in IR β

expression and/or INS signaling present in other cellular populations within the hypothalamus. This phenomenon could potentially contribute to the elevated INS sensitivity noted in females¹³.

Moreover, for a comprehensive evaluation of IR recombination efficiency, we employed Rosa26 ACTB-tTomato/EGFP reporter mice and quantified the co-localization of TOMATO and GFP post-tamoxifen (TM) treatment in both male and female *RcKO* mice (**Supplementary Figure 2A**). Control mice exhibited approximately 15% Tomato-positive astrocytes in the SCN and 18% in the ARC, without significant sex-related differences. Following TM treatment, both sexes displayed approximately 70% recombination in the SCN and 60% in the ARC, highlighting consistent rates across sexes (**Supplementary Figure 2A**).

These additional findings and discussions have been incorporated into our revised manuscript, enriching the understanding of potential sex-specific impacts of INS signaling on circadian regulation and metabolic homeostasis.

Reviewer's major concern 6: The main concern of this reviewer is related to the interpretation of the sexual dimorphic effect. Several methodological aspects need to be clarified here.

- Was female estrous cycle checked or synchronized? It is relevant to explain how the housing of males and females was done during the running-wheel experiment, were females exposed to male's odor? It has been shown that female's light entrainment changes over the estrous cycle and that the cycle can be influenced by the presence of male's odor. Knowing these methodological details are critical for data interpretation specially because it is not clear from the text and from the figure legend whether the daily pattern of locomotor activity was calculated as an average of one or several days on each light condition.

- It is also important to know whether the light intensity to which males and females were exposed to during LD experiments was the comparable. One could see from the representative actograms that the offset of activity in males and females is different but without this methodological detail it is difficult to attribute this effect to a sex dimorphism.

Response: Thank you for raising important methodological questions that may influence our results and interpretations. We have now included information about the housing conditions and the calculation of the daily pattern of locomotor activity in the "Methods" section, specifically under the subsection "Circadian locomotor activity" (**Pages 24-25**). Through these clarifications, we aim to provide comprehensive responses to your questions, thereby enhancing the overall coherence and comprehension of our manuscript and ensuring the utmost rigor and reproducibility within our experimental framework.

Regarding the housing conditions during the running-wheel experiment, both male and female mice were individually housed in separate ventilated cages located in a barrier area, with each cage equipped with its own running wheel. To prevent any potential interference between males and females, mice were placed in separate ventilated racks, entirely preventing direct physical or olfactory engagement between the two groups. This approach was meticulously executed to minimize the exposure of females to male odors and eliminating any potential influence of male presence on female circadian entrainment.

To calculate the daily pattern of locomotor activity, we analyzed the wheel-running data for each mouse over a span of at least 7-10 days. This extended timeframe allowed us to comprehensively define the overall daily pattern of locomotor activity. By doing so, we effectively accounted for potential changes related to the estrous cycle, ensuring a robust representation of the circadian entrainment pattern in both male and female mice.

Regarding lighting conditions, we can confirm that the light intensity to which males and females were exposed during LD experiments was the same. These experiments were conducted in the same room and in similar positions in the racks, thus ensuring consistent and uniform lighting conditions for all animals.

Reviewer's major concern 7: For the data shown in Fig 3, authors should indicate the way in which they administered Insulin in the FFA shift experiment. It is also important to explain the reasoning behind choosing ZT12 as the time for INS injection. This is relevant to understand whether the effect of the acute manipulation of INS levels FAA phase is-dependent.

Response: Thank you for your valuable feedback and for emphasizing the need for clarity regarding the administration of insulin in the FFA shift experiment and the rationale behind selecting ZT12 for INS injection in Figure 3. We apologize for any confusion and have now addressed this concern in the manuscript (**pages 9-10, lines 249-257**). Your feedback has significantly enhanced the manuscript's clarity and comprehensiveness.

Reviewer's major concern 8: The weight and body composition changes shown in Fig 4 could have a growth restriction component? Was this possibility evaluated?

Response: Thank you for raising the important concern about potential growth restriction in the mutants. We've carefully addressed this issue and can provide further clarity on this crucial aspect. We monitored the growth of the mutants exposed to a 60% high-fat diet (HFD) over a 20-week period, as illustrated in the **new Figure 6** and **supplementary Figure 6**. We observed a marked reduction in growth even within the context of a HFD, as

depicted in the new **Figure 6B**. This finding underscores the significant impact of IR deletion on growth regulation. Additionally, it's worth noting that certain studies have hinted at the possibility of a compensatory up-regulation of the insulin-like growth factor 1 receptor (IGF1-R) when the IR is ablated in astrocytes¹⁴. Despite the presence of elevated serum IGF1 levels (**Supplementary Figure 6A**), male *IRcKO* mice continued to display impaired growth. This observation underscores the overriding influence of IR loss in astrocytes on growth regulation, potentially outweighing any compensatory effects mediated by IGF1. We deeply value the insightful feedback provided by the reviewer, as it offers valuable context that enriches the broader implications of our study.

Reviewer's major concern 9: In Fig 5 authors show data from an alternative genetic manipulation of IR in astrocytes using a viral approach to induce the knock-down, now more specifically in the VMH. In order to interpret the results properly a validation of the AAV approach is necessary.

Response: We appreciate your concern regarding the validation of the AAV approach utilized to knock down the IR in astrocytes, particularly within the ventromedial hypothalamus. We recognize the critical importance of establishing the effectiveness and specificity of the viral-mediated knockdown technique to ensure accurate result interpretation. In response to this concern, we have implemented significant revisions in our manuscript.

Firstly, to provide visual affirmation of the specificity and success of our knockout approach, we have included representative images displaying GFP fluorescence. These images elucidate the extent of AAV delivery and astrocytic transduction within the mediobasal hypothalamus (MBH), as seen in **Figure 7A**. These visual depictions serve to confirm the efficient AAV delivery and the successful transduction of astrocytes within the designated hypothalamic nuclei, thereby reinforcing the accuracy of our knockout strategy.

Secondly, as a response to Major Concern #4 from Reviewer 2, we have conducted additional experiments aimed at knocking out astrocytic IR in both the ventromedial nucleus (VMH) and the arcuate nucleus (ARC).

Additionally, we have updated the terminology for the ventromedial hypothalamus, now referred to as the mediobasal hypothalamus (MBH = arcuate + ventromedial nuclei), as reflected in the revised manuscript.

Notably, our new findings indicate that in males, other brain regions and circuits beyond the MBH contribute to metabolic and circadian phenotypes. In contrast, in females, astrocytic INS signaling within the ARC emerges as pivotal for metabolic and circadian phenotypes (**Figure 7 and supplementary Figure 7**). The inclusion of this data substantiates the credibility of our experimental model and bolsters the reliability of our findings' interpretation. Furthermore, it underscores the sex-specific roles that hypothalamic astrocytic INS signaling plays in governing metabolic and circadian processes.

Reviewer's major concern 10: Again, in Fig 5 it is not clear from the text and from the figure legend whether the daily pattern of locomotor activity was calculated as an average of one or several days on each light condition.

Response: We greatly appreciate your meticulous attention to detail regarding the calculation of the daily pattern of locomotor activity, particularly in the context of **new Figure 7**. To provide clarity, the determination of the daily pattern of locomotor activity involved the analysis of wheel-running data gathered over a period of no less than 7-10 days for each individual mouse. This extended timeframe was deliberately chosen to ensure a comprehensive and robust representation of the overall circadian entrainment pattern, as highlighted in our response to Major Concern 6. To address this concern, we have included this specific information within the legend of **Figure 7**, as well as in the "Methods" section, specifically under the subsection "Circadian locomotor activity" (**Pages 24-25**).

Reviewer's major concern 11: Authors should also comment on the very reduced locomotor activity observed in females after the viral manipulation to delete astrocytic IR in the MBH of the females (Fig 5C).

Response: Thank you for your thorough review and insightful observations. Your concern about the reduced locomotor activity in female mice after viral manipulation, as shown in the **updated Figure 7D** (previously Figure 5C), is of great value. We fully acknowledge the need to comprehensively address this observation to shed light on its implications within the wider scope of our study.

The MBH, encompassing the arcuate nucleus (ARC) and ventromedial nucleus (VMH), has been linked to disturbances in circadian locomotion rhythms¹⁵. The decrease in female locomotion aligns with the suppressed nocturnal activity observed in female *IRcKO* mice. Interestingly, our targeted AAV injections to ablate IRs within ARC astrocytes, conducted in response to Referee #2's Major Concern 4, yielded similar outcomes, displaying significant reductions in female locomotor activity (**New Figure 7I**). This emphasizes the role of ARC astrocytic IR in regulating female locomotion. These findings closely match earlier research highlighting the importance of ARC-specific leptin-sensitive neurons in controlling daily locomotion rhythms¹⁶⁻²¹. Interestingly, the absence of circadian variation in leptin receptor expression within these neurons²² implies potential modulation by SCN's inputs. This concept is supported from the established role of the direct SCN-ARC connection in shaping locomotor patterns²³. In this context, it's conceivable that the deletion of IRs in ARC astrocytes could reshape the leptin-sensitive network, potentially influencing entrainment to photic cues and/or the synchronization of ARC

neuronal circuits. We have thoughtfully incorporated these insights into the revised “Discussion” section (**new page 21, lines 607-620**) of our manuscript to provide a clearer elucidation of the implications stemming from the observed decrease in female locomotion within IR-MBH-KO and IR-ARC-KO mice.

Your meticulous feedback has significantly enriched the depth and coherence of our research, and we extend our sincere gratitude for your invaluable contribution.

Reviewer’s major concern 12: Please, indicate food access period in panel E (RF 6h and RF 4h).

Response: We appreciate your valuable feedback concerning Figure 5E (new **Supplementary Figure 7H**), and the need to indicate the food access period. We apologize for the oversight in not explicitly providing this information. In direct response to your observation, we have explicitly labelled and detailed the food access period within the figure.

Reviewer’s major concern 13: The involvement of dopamine levels explaining how the deficit of IR in astrocytes alters INS-induced entrainment might be stronger if authors really confirm that dopamine levels are reduced in their IRcKO. Moreover, authors should indicate the way in which they administered DRD2 agonist.

Response: We appreciate your valuable feedback regarding the role of dopamine (DA) levels and the administration of the DRD2 agonist quinpirole in our study. Your insights have prompted us to conduct further investigations and provide clarifications, resulting in significant enhancements to our manuscript. Specifically, we assessed DA levels and its metabolites in both the striatum and hypothalamus of control and *IRcKO* mice using HPLC (**New Figure 4A and 4B**). The results revealed distinct changes in the striatum, characterized by reduced DOPAC levels and subsequent reductions in the ratios of DOPAC/DA and DOPAC + HVA/DA (**Figure 4A**). Importantly, the levels of DA itself remained unchanged, indicating that the alteration is related to DA release dynamics rather than DA synthesis. In contrast, the hypothalamus displayed notably higher levels of DA, DOPAC, and noradrenaline (NA), accompanied by increased DOPAC/DA ratio (**Figure 4B**). These findings suggest that the altered DA release primarily occurs within the striatum and is not indicative of a general defect in neurosecretory granule release or turnover. This observation is in line with previous research involving brain-specific IR knockout mice²⁴. Taken together with the results obtained from the administration of the DRD2 agonist, these findings imply that changes in DA turnover within specific brain regions may contribute to the observed alterations in circadian locomotor activity and FAA in *IRcKO* mice.

On the other hand, we apologize for any confusion arising from the initial presentation of information about the administration of the DRD2 agonist quinpirole in the manuscript. We understand the significance of clearly detailing the administration method and dosage. The DRD2 agonist quinpirole (1 mg/kg body weight) was administered intraperitoneally to the animals daily at ZT12 for a duration of 5 days. This method of administration and dosage were selected based on previous studies and its known effectiveness in modulating DRD2 activity in mice^{25,26}. We have revised the methods (under the epigraph “Animals and Treatments” on **page 23, lines 663-666**) and results (on **Page 11, Lines 305**) sections to explicitly provide these details. We believe that this clarification strengthens the transparency and rigor of our study.

MINOR CONCERNS

Reviewer’s minor concern 1: Page 2 line 32, abbreviation for INS has to be explained the first time is mentioned.

Response: We apologize for not explaining the abbreviation for “INS” (insulin) when it was first mentioned. In the revised version of the manuscript, we have provided the full name “insulin” along with the abbreviation “INS”. We have also ensured that abbreviations are not used in the abstract, and the term “insulin” is introduced on **page 3, line 71**.

Reviewer’s minor concern 2: Page 3 line 57: One can argue that shift work not only induces mistimed feeding but also mistimed light exposure and both might disrupt circadian physiology.

Response: This is an excellent point. We appreciate the reviewer’s point about the potential impact of mistimed light exposure in addition to mistimed feeding in shift work. In the revised version of the manuscript, we have acknowledged that shift work can disrupt circadian physiology not only through mistimed feeding but also through mistimed light exposure (**line 55 of the new version of the manuscript**).

Reviewer’s minor concern 3: Page 3 lines 69-71: The fact that 20% of oscillating proteins do not show signs of rhythmicity at mRNA level suggests that rhythmic regulation of transcription is just one aspect but post-transcriptional, translational, post-translational protein stability, subcellular localization are also regulated rhythmically by the molecular clock.

Response: This is again, a very good point. In the revised version of the manuscript, we have expanded on this point, emphasizing that rhythmic regulation occurs not only at the transcriptional level but also at post-transcriptional, translational, and post-translational levels, as well as in protein stability and subcellular localization (**lines 69-71 of the new version of the manuscript**). We appreciate the reviewer’s valuable input.

Reviewer's minor concern 4: Page 3 line 71: the word clock seems to be missing after molecular

Response: We apologize for the omission of the word "clock" in the sentence. In the revised version of the manuscript, we have added the word "clock" after "molecular" to accurately reflect the intended meaning (**line 69 of the new version of the manuscript**).

Reviewer's minor concern 5: Page 3 line 75: circadian clock has to be deleted

Response: We apologize for the incorrect use of the term "circadian clock" in the sentence. In the revised version of the manuscript, we have removed the words "circadian clock" (**line 74 of the new version of the manuscript**).

Reviewer's minor concerns 6 and 9: Page 4 line 90 and Page 5 line 130: *in vitro* and *in vivo* goes in italics

Response: Thank you for your observation. We have made the appropriate changes by using italics for "*in vitro*" and "*in vivo*".

Reviewer's minor concern 7: Page 5 line 120: "astrocyte clock" has to be changed by "astrocyte's clock"

Response: We appreciate the reviewer's suggestion to change "astrocyte clock" to "astrocyte's clock" for better grammatical accuracy. In the revised version of the manuscript (**line 123 of the new version of the manuscript**), we have made this correction to ensure clarity and proper phrasing. Thank you for bringing this issue.

Reviewer's minor concern 8: Page 5 line 126: "onset of increase feeding" has to be changed by "onset of activity"
Response: We have changed "onset of increase feeding" to "onset of activity" in the new version of the manuscript.

Response: Thank you for noting the more accurate description of "onset of activity" instead of "onset of increased feeding." We have made this correction in the revised version of the manuscript to reflect the intended meaning (**line 130 of the new version of the manuscript**).

Reviewer's minor concern 10: Page 6 line 165: IRcKO female and male?

Response: Thank you for pointing out this omission. Including both sexes in our study is important for a comprehensive understanding of the effects of astrocyte specific *IRcKO* mice. We apologize for any confusion caused by the previous version of the manuscript, and we have rectified this oversight in the revised version (**line 170 of the new version of the manuscript**).

Reviewer's minor concern 11: Page 6 line 169: *Zeitgeber* time goes in italics

Response: We appreciate the reviewer's feedback and used italics for "*Zeitgeber*" in the revised version of the manuscript.

Reviewer's minor concern 12: Page 6 line 174: The sentence has to be re-written, as a suggestion: To uncover the contribution of astrocyte's INS signaling on the regulation of circadian locomotor activity.

Response: Thank you for providing a suggestion to rephrase the sentence for clarity. We have followed your suggestion and rephrased the sentence to read: "*To uncover the contribution of astrocyte's INS signaling on the regulation of circadian locomotor activity.*" (**Lines 192-193 of the new version of the manuscript**).

Reviewer's minor concern 13: Page 7 line 206: did authors mean male and female mutants here?

Response: We apologize for the mistake in the sentence you pointed out on **page 9, line 241 of the new version of the manuscript**. We have corrected the sentence to clarify that the observed reduction in daily activity in the dark period applies to both male and female controls and mutants. The revised sentence now reads as follows: '*...and was associated with a reduction of daily activity in the dark period in both controls and mutants, regardless of sex.*'

Reviewer's minor concern 14: Page 8 line 219-220: should it read "...this effect was absent in IRcKO female mice"?

Response: Regarding your concern about **page 9, lines 257-260 of the revised version of the manuscript**, we confirm that the original text is correct. The sentence states, "*Remarkably, our results indicate that INS and glucose can acutely shift the rhythm and duration of FAA, and this effect was absent in IRcKO male mice (Figure 3B, C).*" The subsequent sentence then discusses the impact on females, stating that "*Surprisingly, in females, INS and glucose did not impact the duration or phase of the FAA in both groups of mice (Figure 3B, C).*" Thus, we have accurately conveyed that the effect of INS and glucose manipulation was absent in *IRcKO* male mice, while it did not impact the FAA in females of both control and *IRcKO* groups. We appreciate your careful review and apologize for any confusion caused by the sentence.

Reviewer's minor concern 15: Page 8 line 232: locomotor rhythms sounds more accurate than behavioral rhythms here.

Response: Regarding your concern on **new page 9, line 269**, we agree that using the term '*locomotor rhythms*' would be more accurate than '*behavioral rhythms*' in that specific context. We appreciate your keen attention to detail and have made the necessary revision to ensure the precision and accuracy of our manuscript.

Reviewer's minor concern 16: Page 10 line 276: The sentence has to be re-written, as a suggestion: ...the role of VMH astrocytes's INS signaling in sustaining...

Response: Thank you for providing a suggestion to rephrase the sentence (**page 17, line 490 of the new version of the manuscript**). In the revised version of the manuscript, we have rephrased the sentence as you suggested.

Reviewer's minor concern 17: Page 11 line 315: The word "astrocytes" is repeated 3 times in the same sentence

Response: We apologize for the repetition of the word "astrocytes" in the sentence. In the revised version of the manuscript, we have revised the sentence to eliminate the repetitive use of the term (**new page 10, lines 278-280**): *"It has been reported that the deletion of IR on astrocytes within the nucleus accumbens of the striatum leads to a decrease in ATP exocytosis, resulting in reduced purinergic signaling on dopaminergic neurons"*.

Reviewer's minor concern 18: Page 11 line 318: the word "and" y repeated twice

Response: We apologize for the repeated word "and" in the sentence. In the revised version of the manuscript, (**page 10, lines 282-283**) we have made the necessary adjustment to eliminate the redundancy and improve the sentence structure (*"Notably, DA plays a crucial role in feeding, locomotor behavior, as well as photic and food entrainment"*).

Reviewer's minor concern 19: Page 13 line 372: *in vitro* goes in italics

Response: We have made the correction and used italics for "*in vitro*" in the revised version of the manuscript.

Reviewer's minor concern 20: Page 14 lines 414: *ad libitum* goes in italics

Response: Thank you for pointing out the need for italics in the term "*ad libitum*" In the revised version of the manuscript, we have applied italics to "ad libitum" to conform to proper formatting.

We sincerely thank the reviewer for providing these valuable suggestions and feedback. Your input has greatly contributed to the overall quality and clarity of our manuscript. We have carefully addressed each of the concerns raised, and we believe the revised version of the manuscript now reflects these improvements. We appreciate your time and effort in reviewing our work, and we look forward to your evaluation of the revised manuscript. Thank you very much.

REVIEWER#2

Overall comment: In the present manuscript Gonzalez Vila et al. perform a solid and deductive series of experiments to demonstrate that a limiting factor for the control of circadian behavior is the role of the insulin receptor in astrocytes. This regulation occurs in a sexually dimorphic manner and is mediated by dopamine signaling. Identifying this critical role in astrocytes is fundamental to further understand components of one of the most complex and most essential processes for survival, food intake. Given the exponential rise in obesity and pathologies associated to energy balance dysregulations, it is critical to further understand in a holistic manner and including all the pieces of the bigger picture machinery the feeding process. This manuscript brings us a step closer in understanding how our body responds to temporal shifts in feeding regulation which frequently happen in the human population.

The manuscript is solid, data is adequately analyzed following appropriate statistic tests and is of interest to a broad readership. Taken together the strength of these new sets of data, together with the scope of the journal I believe this manuscript is a fit for Nature Communications.

However, a set of suggestions listed below could elevate the scope and strengthen this manuscript. Most of those points could be answered by expanding the present discussion.

Response: We would like to express our heartfelt gratitude to Reviewer#2 for dedicating their time and effort to a comprehensive review of our article. His/her encouraging comments and insightful recommendations further strengthen our study. In response to the reviewer's invaluable feedback, we undertook additional experiments and conducted a meticulous revision of the original manuscript. The reviewer's constructive input has played a crucial role in clarifying data interpretation and elevating the overall standard of the paper. We deeply appreciate their contribution to the advancement of our research.

MAJOR CONCERNS

Reviewer's major concern 1: Authors demonstrate that insulin plays a fundamental role in regulating circadian rhythmicity in astrocytes. One conceptual question that can be raised is: does the concentration of insulin in the brain oscillate in a daily manner? Is there a link between obesity (or associated pathologies) and the leakage of the blood brain barrier, consequently affecting the concentration of insulin in the brain? Is astrocyte insulin signaling impaired upon obesity due to insulin resistance?

Response: We appreciate the reviewer's insightful questions regarding the interplay between brain INS concentration, blood-brain barrier (BBB) integrity, and astrocyte INS signaling, particularly in the context of

obesity. These questions have motivated us to further investigate these intricate relationships and enhance the discussion of these concepts in our revised manuscript.

Addressing the query about daily fluctuations in brain INS concentration, existing literature suggests a correlation between plasma INS levels and cerebrospinal fluid (CSF) INS levels, indicating potential diurnal oscillations^{27–30}. These oscillations align with the observed 24-hour rhythm in brain glucose utilization, suggesting a connection between INS dynamics and brain metabolic activity^{1,2}. We have now investigated the diurnal expression patterns of IR, glucose transporters, and INS signaling within the hypothalamus at various times of the day (please see our response to Referee #1's Major Concern 1). These findings, coupled with our transcriptome analysis of synchronized astrocytes and reported diurnal INS oscillations in the brain, provide compelling evidence suggesting astrocytes' potential contribution to the diurnal regulation of hypothalamic INS sensitivity and/or glucose availability.

In the context of obesity, previous studies have highlighted the adverse effects of a high-fat diet (HFD), including neuronal loss and BBB disruption in rodent models and humans³¹. These findings underscore the crucial role of astrocytes in maintaining BBB integrity and responding to HFD challenges. Our novel results with *IRcKO* mice offer compelling insights into this relationship. We observed heightened hypothalamic INS signaling upon HFD exposure, accompanied by resistance to diet-induced obesity (DIO), as shown in **new Figure 6 and Supplementary Figure 6**. The increased INS signaling in mutant mice could be a compensatory response due to astrocytic IR deficiency, potentially protecting against adverse effects of HFD and contributing to DIO resistance. These observations suggest a protective role of astrocytic INS signaling against DIO development. Moreover, the altered metabolic environment during DIO might lead to adaptations in astrocytic INS responses, disrupting regular regulatory pathways. This could result in astrocytes actively contributing to mediating obesity-related metabolic consequences through INS responses. These insights position astrocytes as potential key players in the intricate interplay between metabolic regulation, obesity, and the brain, offering a novel avenue for exploration. The complex interplay between obesity, BBB integrity, INS dynamics, and astrocyte signaling remains an ongoing area of research. We greatly value the reviewer's insightful input and are committed to incorporating these considerations into our ongoing and future research efforts.

Reviewer's major concern 2: Diet induced obesity attenuates FAA in mice both in HFD and OB/OB mice. Given that obese subjects are both leptin resistant and insulin resistant, I was wondering whether authors have explored the possibility of leptin signaling being involved in the fundamental role of astrocytes in circadian behavior. For instance, do *IRcKO* males and females change their entrainment to feeding time when chronically fed a HFD? Is the phenotype still sexually dimorphic?

Response: We greatly appreciate your valuable feedback and insightful consideration of the potential interplay between leptin signaling, INS resistance, and astrocytic INS signaling in the context of diet-induced obesity (DIO).

Regarding the influence of leptin signaling on astrocytic INS-mediated circadian behavior, we investigated female and male *IRcKO* mice exposed to a 60% high-fat diet (HFD) for 20 weeks (**Figure 6H and Supplementary Figure 6H-6K**). Mutants showed increased locomotor activity under HFD, whereas their activity patterns resembled controls under a standard diet (STD) for males and were reduced for females. This suggests that astrocytic INS signaling may differentially impact circadian locomotor activity based on dietary composition. During HFD restricted feeding, male *IRcKO* mice displayed enhanced food-anticipatory activity (FAA) (**Supplementary Figure 6H-6K**), indicating an interplay between astrocytic INS signaling and HFD effects on FAA. With leptin resistance in obesity, our results reflect a complex relationship involving astrocytic INS signaling, leptin levels, and metabolic adaptation, warranting further exploration.

Importantly, the sexually dimorphic responses in leptin levels and metabolic phenotypes between male and female *IRcKO* mice highlight the complexity. Males exhibited reduced leptin levels under both STD and HFD (**Supplementary Figures 5A and 6A**), while females showed no significant changes in leptin levels in any conditions (**Supplementary Figures 5A, 6F**), possibly due to sex-specific differences in leptin regulation influenced by hormonal changes and fat distribution^{32–35}. These variations contribute to contrasting metabolic adaptations between male and female mutants, emphasizing the importance of exploring specific neuronal circuits and leptin receptor-expressing neurons for understanding the role of astrocytic INS signaling in FAA and metabolic regulation.

We value your insightful questions that have expanded our research dimensions.

Reviewer's major concern 3: The reviewer wants to elevate the fact that authors use ANCOVA and point to the regression curves when analyzing energy expenditure. It is an important point that should be seen in all studies doing indirect calorimetry.

Response: We appreciate the reviewer's acknowledgement of our use of ANCOVA and regression curves in energy expenditure analysis. These statistical methods indeed enhance the strength of our variable relationships and allow for necessary adjustments and comparisons. To ensure clarity, in the revised manuscript, we have

emphasized and discussed the interpretation of the regression curves in relation to energy expenditure analysis (page 13, lines 373-377).

Reviewer's major concern 4: Authors elegantly show that IR in the VMH astrocytes (including VMN and ARC) is important for circadian behavior. Could authors narrow down between the VMN and the ARC which is the major contributing cluster. Would the close proximity to a circumventricular organ be a key factor allowing broader sensing due to capillary fenestration and better sensing. Building up on this question, as a discussion point authors could consider suggesting a potential role of astrocytic IR in circumventricular organs being an important part of circadian rhythmicity. Other circumventricular organs such as AP or SFO could be evaluated/discussed.

Response: We sincerely appreciate the thoughtful feedback provided by the referee and the insightful questions raised. Your comments have been invaluable in refining our research and enriching the depth of our study.

Regarding the major contributing cluster between the ventromedial nucleus (VMH) and the arcuate nucleus (ARC), we have conducted additional experiments to gain deeper insights into their specific roles in circadian behavior and metabolic regulation (**new Figure 7 and Supplementary Figure 7**). We found that male mice with IR deletion in MBH astrocytes, whether in the VMH or the ARC, did not exhibit significant differences in metabolic or locomotor parameters that resemble the metabolic and circadian phenotypes observed in *IRcKO* mice. This suggests that while mediobasal astrocytic INS signaling indeed contributes to controlling food intake and energy expenditure in males, other brain regions and circuits likely play pivotal roles in shaping the broader metabolic and locomotor phenotype exhibited by *IRcKO* males. Interestingly, our findings diverge in females, where deleting IR in VMH astrocytes resulted in an increase in the amplitude of the periodogram, a contrast to the phenotypes seen in *IRcKO* mice. **In female mice with IR deletion specifically in the ARC (IR-ARC-KO), both metabolic and circadian locomotor alterations closely resembled those observed in *IRcKO* females.** These alterations included reduced body weight, cumulative food intake, and locomotor activity, alongside increased energy expenditure. These outcomes underscore the critical role of astrocytic INS signaling within specific hypothalamic nuclei, particularly the ARC, in orchestrating metabolic homeostasis and circadian locomotor behavior.

Your suggestion regarding the potential influence of astrocytic IR within circumventricular organs (CVOs) on circadian rhythmicity is greatly appreciated. The enriched expression of IRs within astrocytes in CVOs¹⁰⁷, coupled with our findings in IR-ARC-KO mice and the established connection between the SCN and ARC known to contribute to shaping locomotor patterns¹⁰⁴, introduces intriguing possibilities for understanding their participation in the regulation of circadian rhythms. For instance, astrocytes within CVOs have been reported to express vasopressin (AVP) receptors¹⁰⁶, suggesting a potential link between astrocytic activities in CVOs and the regulation of circadian rhythms through the modulation of AVP signaling within the SCN. These findings suggest that future investigations of CVO astrocytes in circadian rhythms and their impact on brain function are indeed warranted. We have now broadened the discussion and include the potential link of CVO astrocytes in regulating circadian rhythms (**page 21, lines 620-628** of the revised version of the manuscript).

We extend our gratitude to the Reviewer for providing these valuable suggestions.

Reviewer's major concern 5: Given the important role for dopamine signaling for circadian behavior. Do disorders or alterations in dopamine production impair circadian behavior?

Response: We greatly appreciate your insightful question about the impact of dopamine (DA) signaling on circadian behavior and the potential consequences of disorders or alterations in DA production. Indeed, disruptions or changes in DA production can have significant effects on circadian behavior. For instance, conditions like Parkinson's disease, characterized by reduced DA levels, can lead to impairments in locomotor activity, REM sleep, and clock gene expression. Similarly, imbalances in DA levels, observed in disorders such as cocaine exposure, attention-deficit/hyperactivity disorder, depression, and schizophrenia, can also influence circadian behavior⁴⁰. Moreover, the intricate interplay between the DA system and circadian rhythms highlights a bidirectional relationship. DA receptor can impact clock gene expression, while circadian-associated genes can in turn modulate DA synthesis. For example, mice deficient in brain *Bmal1* or *Per2* genes exhibit mania-like behavior, whereas mice lacking both *Cry1* and *Cry2* genes show altered anxiety-like behavior⁴¹. Notably, affective disorders like major depressive disorder and bipolar disorder, often characterized by disrupted circadian rhythms, further emphasize the connection between DA and circadian regulation⁴⁰.

We find your inquiry particularly insightful, as it adds depth to our understanding of the complex interactions between neurotransmitter systems and circadian behavior.

Reviewer's major concern 6: Given the sexually dimorphic phenotype, can authors expand on the origin of such dimorphism. Are there sexual hormonal regulation associated to mechanism identified?

Response: We appreciate the Reviewer's attention to the sexually dimorphic phenotype observed in our study and the potential influence of sexual hormonal regulation on the identified mechanisms. In response to your inquiry, we conducted bilateral ovariectomy (OVX) and sham operations on both control and *IRcKO* female mice.

Our investigation of the circadian phenotype revealed that both sham operated and OVX female mutant mice displayed reduced activity during the dark phase compared to control animals, accompanied by a decrease in the amplitude of the periodogram (**New Figure 2C and 2D**). These findings suggest that ovarian function may not be a primary determinant of the altered locomotor phenotype observed in female *IRcKO* mice. This aligns with previous studies indicating that estrogens are not essential for maintaining circadian rhythms⁴². By eliminating the direct influence of ovarian hormones through OVX, these findings further underscore the potential role of INS signaling in astrocytes as a mediator of sex-specific differences in the regulation of circadian locomotor activity.

Regarding the metabolic phenotype, we present comprehensive results in **New Figures 5G-I and Supplementary Figures 5D-F**, indicating that the dimorphism observed in body weight and composition, food intake, and glucose homeostasis between *IRcKO* mice and controls is indeed contingent on ovarian function. These results provide valuable insights into the interplay between astrocytic INS signaling, ovarian hormones, and metabolic regulation.

MINOR CONCERNS

Reviewer's minor concern 1: We recommend a thorough Spell-check identified before resubmission. Some examples identified in the text.

Line 33: Meaning of INS in the abstract should be detailed

Line 143: When they say figure C, I guess authors refer to figure 1C right.

Line 551: Posthoc instead of postdoc

Response: We sincerely appreciate the thoroughness of the reviewer in identifying these minor issues within our manuscript. All the typos have been corrected.

We extend our sincere gratitude to the Reviewer for offering invaluable suggestions and feedback. Your insightful input has significantly enhanced the quality and lucidity of our manuscript. We have thoughtfully addressed each of the concerns you raised, and we are confident that the revised version of the manuscript now embodies these enhancements. We truly appreciate the time and dedication you invested in reviewing our work, and we eagerly anticipate your assessment of the revised manuscript. Thank you sincerely for your invaluable contribution.

REVIEWER#3

Overall comment: Recent work indicates that insulin and dopamine signaling is involved in feeding entrainment of daily rhythms. Here the authors test the role of insulin signaling in astrocytes and whether this is necessary for entrainment of daily rhythms by light and feeding. They find that insulin is sufficient to reset the astrocyte clock, with effects that differ from those induced by glucocorticoids. They also find that eliminating IR in astrocytes modulates activity levels and circadian period in females. Eliminating IR in astrocytes in adulthood also increases FAA acquisition in each sex, but at different phases of food entrainment. Changes in metabolism were also detected after deletion of IR in astrocytes, and these effects were influenced by sex. VMH-specific IR astrocyte deletion altered activity levels and metabolism. DrD2 agonists modulated activity levels. Overall, this work suggests that IR signaling in astrocytes modulates activity levels and metabolism.

Response: We sincerely appreciate Reviewer #3 for their thoughtful comments and constructive feedback regarding our manuscript. His/her insightful input has played a pivotal role in refining the quality and coherence of our work. The outcomes of these additional experiments have further substantiated the findings and conclusions of our study. We also believe that these improvements have bolstered the content of the paper and added to its scientific significance.

MAJOR CONCERNS

Reviewer's major concern 1. The individual experiments are technically sound, but the results are not noteworthy. This work will make a larger impact if the causal connections between findings are more clearly articulated.

Response: We highly appreciate Reviewer #3's insightful feedback, particularly their emphasis on enhancing the clarity of causal connections within our work. To address this concern, we have implemented significant improvements in our manuscript. By weaving contextual information, relevant background knowledge, and explanatory passages, we aim to provide a clearer understanding of the cause-and-effect relationships underlying our experimental outcomes. These additions are designed to illuminate how each aspect of our research contributes to the broader conceptual framework of insulin signaling in astrocytes and its implications for daily rhythms and metabolism.

Moreover, we have undertaken additional experiments to further solidify the mechanistic basis of our study:

1. We present evidence illustrating the diurnal regulation of hypothalamic insulin receptor sensitivity (**Supplementary Figures 1E and 1F**).
2. We investigated dopamine levels in the striatum and hypothalamus of control and mutant mice (**Figure 4A and 4B**).
3. We explored sexual dimorphic phenotypes through ovariectomies, analyzing metabolic and circadian aspects (**Figures 2 and 5 and Supplementary Figures 2 and 5**).
4. We examined the metabolic and circadian phenotypes of mutants under high-fat diet conditions (**Figure 6 and Supplementary Figure 6**).
5. We performed additional stereotactic injections targeting specific hypothalamic nuclei to validate the role of astrocytic IR in energy balance and circadian behavior (**Figure 7 and Supplementary Figure 7**).

Together, these enhancements, combined with the improved narrative in the manuscript, strengthen the articulation of causal connections between our findings. We believe that these revisions offer readers a more comprehensive and insightful understanding of the implications of our study. We are confident that these efforts have effectively addressed Reviewer #3's concern and have resulted in a more coherent and impactful manuscript.

Reviewer's major concern 2: In several places, the authors make inferences that are not clearly supported by evidence. For instance, effects of IR signaling in astrocytes may impact locomotor levels and metabolism by acting through non-SCN circuits, rather than reflecting changes in the SCN neural network (as concluded on lines 188-189, 291-292, 305-306, 310-311, 327). Alternative interpretations are not considered or discussed by the authors. In addition, the authors state that "Our results first suggest that astrocytes are part of the FEO". The current data deserve a more nuanced interpretation and this claim requires a higher bar of proof than provided by the current work.

Response: We greatly value Reviewer #2's insightful feedback, particularly their concern about potential inferences lacking sufficient evidence in our manuscript. We have carefully addressed this concern by implementing comprehensive revisions to our text.

Upon a thorough reevaluation of the identified sections (**new lines 212-214, 518-521, 528-534, 546-550 of the revised version of the manuscript**), we have adopted a more balanced and nuanced approach. Moreover, we now explicitly acknowledge alternative interpretations and delve into potential mechanisms beyond the SCN neural network when discussing the impact of IR signaling in astrocytes on locomotor activity (**lines 228-230**: "While the impact of INS signaling could potentially extend beyond the SCN and influence locomotor activity by affecting circuits outside of it, our findings underscore the role of INS signaling in astrocytes in mediating sex-specific differences in circadian locomotor activity"). By avoiding overgeneralization and refraining from solely attributing effects to the SCN circuit, we aim to present a more accurate representation of potential underlying mechanisms.

Furthermore, we appreciate the Reviewer's point regarding our claim about astrocytes' involvement in the FEO. Recognizing the need for a more cautious interpretation, we have revised our statement to convey a more measured understanding of our findings. Specifically, we now explicitly state that while our results suggest a potential role for astrocytes in the FEO, definitive confirmation requires additional experimental approaches. We acknowledge the intricate nature of circadian behavior and emphasize the necessity for supplementary evidence to conclusively establish astrocytes' direct contribution to the FEO. To this end, we have enriched our discussion by outlining future research avenues that could provide more robust evidence, such as exploration of astrocyte-neuron interactions within FEO-relevant brain regions, including the striatum and/or the hypothalamus (**page 19, lines 569-572 of the new version of the manuscript**). In conclusion, we approach our discussion with heightened caution, underscoring the need for further substantiation of astrocytes' role in the FEO. These revisions aim to enhance the scientific rigor of our conclusions and contribute to advancing our comprehension of circadian behavior and food anticipation.

We extend our sincere gratitude for the thorough review and constructive feedback, which have been instrumental in elevating the quality and clarity of our manuscript.

Reviewer's major concern 3: The DRD2 agonists do not clearly rescue circadian behavior in IRcKO mice. Instead, this drug had effects in both control and cKO mice.

Response: We deeply appreciate your insightful observation regarding the effects of the DRD2 agonist quinpirole on circadian behavior in *IRcKO* mice. In response to your concern, as well as in addressing the major concern raised by Reviewer #1, we conducted an in-depth analysis of dopamine levels and its metabolites (3,4-dihydroxyphenylacetic acid and homovanillic acid), along with noradrenaline and serotonin along with its metabolite 5-hydroxyindolacetic acid, in both the striatum and hypothalamus of control and *IRcKO* mice using HPLC (**New Figure 4A and 4B**). This analysis revealed distinctive shifts in dopaminergic signaling dynamics across

regions; heightened DA synthesis and turnover were evident in the hypothalamus, while disruptions in DA release were observed in the striatum.

This, together with our investigation into the effects of the DRD2 agonist quinpirole on food anticipatory activity (FAA) rhythms yielded valuable insights into the interplay between INS and dopaminergic signaling in the regulation of FAA in males. The results highlighted substantial shifts in both the rhythm and duration of FAA following quinpirole treatment, seen in both control and mutant animals. Notably, the responses of control animals to quinpirole treatment mirrored those induced by INS shifts. In contrast, these shifts were noticeably impaired in *IRcKO* mutant animals, underscoring the specificity of INS signaling's role in modulating this aspect of circadian behavior (**Figure 4E**). Moreover, in females, while the DRD2 agonist did affect control mice, its short-term administration led to a sustained (1-week) normalization of locomotor activity in mutants (**pages 11-12, lines 333-335, Figure 4F-4I**).

Collectively, these findings illuminate the intricate relationship between DA and INS signaling within the broader context of the circadian regulatory network. We sincerely thank you for your thorough review, which has significantly enhanced the depth and clarity of our manuscript.

MINOR CONCERNS

Reviewer's minor concern 1: Viral-mediated IR knockdown in VMH was not verified or quantified.

Response: Thank you for raising the concern regarding the validation of the AAV approach used for knockout of the IR in astrocytes within the ventromedial hypothalamus. To address this concern, we have made important additions to our manuscript. Firstly, we have modified the labeling of the ventromedial hypothalamus, now referred to as the mediobasal hypothalamus (MBH), in the new version of the manuscript. Secondly, in response to Reviewer 2's suggestion (major concern #4), we conducted additional experiments to specifically knockout astrocytic IR in the ventromedial nucleus (VMH) and arcuate nucleus (ARC) of the hypothalamus. Furthermore, we have included representative images of the GFP fluorescence to illustrate the successful viral delivery and the spreading of the AAV used in the MBH, VMH and ARC (**Figure 7 and supplementary Figure 7**).

Reviewer's minor concern 2: It is not clear if the authors control for familiar wise comparisons (e.g., paired t-test were performed in Figure 1D, which is not appropriate). The use of ANOVA for r.p.m.s and amplitude data in Figures 2, 5, and 6 also seems odd since these data are typically binned and/or summed.

Response: We deeply appreciate your insightful feedback on our manuscript, and we wish to extend our gratitude for your comprehensive evaluation of the statistical analyses presented in our study.

Regarding Figure 1D, we fully acknowledge your concern about the use of paired t-tests for gene expression comparisons across different time points and treatment groups. In response, we have employed one-way ANOVA to assess the effects of DEX or INS treatments more appropriately on the expression levels of *Per2*, *Cry1*, *Bmal1*, and *Dbp* transcripts at 1, 2, and 4 hours in comparison to the baseline time point (0 hours). To address potential issues related to multiple comparisons, we performed post hoc tests using the Tukey-Kramer method. This approach was selected to ensure that any significant differences identified among specific time points are rigorously validated, considering the possible inflation of Type I errors due to multiple comparisons. We are confident that these modifications enhance the statistical validity of our results.

Moreover, in reference to the data presented in Figures 2, new Figures 4 and 7, we want to clarify that we did indeed utilize a repeated measures ANOVA for the analysis of the activity waveform data. We regret any ambiguity that may have arisen in the previous version of our manuscript. To provide enhanced clarity, we have explicitly elucidated this approach within the Methods section, particularly under the "Statistical Analysis" subsection (**lines 860-864**). Given the longitudinal nature of our study and the repeated measurements collected from the same subjects, the use of repeated measures ANOVA was well-suited to capture and analyze patterns of change across multiple time points within each group.

Your meticulous review has substantially improved the quality and precision of our statistical analyses, and we are grateful for your guidance in ensuring the robustness of our findings.

Reviewer's minor concern 3: It is unclear if activity profiles under DD were corrected for period length (Figure 2B).

Response: We extend our gratitude for your meticulous review of our manuscript, and we value your specific inquiry regarding the activity profiles under constant darkness (DD) as depicted in **new Figure 2A**. We want to clarify that the activity profiles under DD were indeed corrected for varying period lengths. The abscissa units in Figure 2A are presented in circadian time (CT), and the mean activity was computed by averaging the activity within 5-minute bins across each animal's circadian cycle. To provide enhanced clarity on this correction process, we have expanded the description of the methodology used to correct for varying period lengths. This elaboration can be found in the "Circadian Locomotor Activity" section of the "Methods" (**lines 726-731 of the new version**).

of the manuscript) We believe this added information ensures a thorough comprehension of the approach we undertook to address your concern.

Reviewer's minor concern 4: Re-entrainment after an extended DD free-run is difficult to quantify and interpret given individual differences in period.

Response: We greatly appreciate your concern regarding the complexities of quantifying and interpreting re-entrainment after an extended DD free run, especially considering the individual differences in circadian periods. Recognizing the challenges posed by these variations, we have taken your feedback to heart and included a specific statement in the Results section (**Page 7, Lines 201-204** of the revised manuscript). This statement highlights the inherent difficulties in quantification and interpretation that arise due to individual differences in circadian periods during re-entrainment.

We sincerely thank you for your insightful feedback, which has contributed to a clearer and more accurate representation of our findings.

Reviewer's minor concern 5: The authors should use "sex" rather than "gender" when referring to rodents.

Response: We appreciate your observation concerning the terminology used in our manuscript. Upon careful consideration, we have made the necessary revisions to address this concern. Specifically, we have replaced instances of the term "gender" with "sex" throughout the text. This adjustment ensures an accurate representation of the biological distinctions between male and female rodents.

Reviewer's minor concern 6: The authors in several places refer to something as "solid" (e.g., line 339), but it would be preferable for the authors to describe the finding or difference clearly.

Response: We appreciate your attention to detail and valuable feedback. We acknowledge the significance of employing clear and precise language in scientific discourse, especially when describing findings and distinctions. This is always complicated for us, as we are not native English speakers and for some words the Spanish meaning is not the same. We concur that the term "solid" may not have adequately conveyed the precise characteristics of the sexual dimorphisms observed in the dopaminergic system. Considering your concern, we have eliminated the potentially confusing usage of the term "solid" and, instead, substituted it with language that offers a more accurate and explicit depiction of the observed sexual dimorphisms within the context of circadian locomotor rhythms.

We sincerely express our gratitude to Reviewer #3 for their invaluable insights and constructive feedback during the review of our manuscript. Their meticulous evaluation has made a substantial contribution to improving the quality, coherence, and scientific merit of our work. The results obtained from the additional experiments not only validate our conclusions but also enhance the overall content of the paper. We deeply appreciate Reviewer #3 for their thorough assessment and invaluable contribution, which has significantly elevated the quality of our manuscript.

REFERENCES

- Jay, T. M., Jouvett, M. & Des Rosiers, M. H. Local cerebral glucose utilization in the free moving mouse: a comparison during two stages of the activity-rest cycle. *Brain Res.* **342**, 297–306 (1985).
- Room, P. & Tielemans, A. J. P. C. Circadian variations in local cerebral glucose utilization in freely moving rats. *Brain Res.* **505**, 321–325 (1989).
- Rivkees, S. A., Fox, C. A., Jacobson, C. D. & Reppert, S. M. Anatomic and functional development of the suprachiasmatic nuclei in the gray short-tailed opossum. *J. Neurosci.* **8**, 4269–4276 (1988).
- Cassone, V. M. Circadian variation of [¹⁴C]2-deoxyglucose uptake within the suprachiasmatic nucleus of the house sparrow, *Passer domesticus*. *Brain Res.* **459**, 178–182 (1988).
- Rodríguez-Cortés, B. *et al.* Suprachiasmatic nucleus-mediated glucose entry into the arcuate nucleus determines the daily rhythm in blood glycemia. *Curr. Biol.* **32**, 796-805.e4 (2022).
- Simpson, I. A. *et al.* Blood-brain barrier glucose transporter: effects of hypo- and hyperglycemia revisited. *J. Neurochem.* **72**, 238–247 (1999).
- Crosby, P. *et al.* Insulin/IGF-1 Drives PERIOD Synthesis to Entrain Circadian Rhythms with Feeding Time. *Cell* **177**, 896-909.e20 (2019).
- Jiang, X. *et al.* SLC7A14 imports GABA to lysosomes and impairs hepatic insulin sensitivity via inhibiting mTORC2. *Cell Rep.* **42**, (2023).
- Eizirik, D. L., Cardozo, A. K. & Cnop, M. The role for endoplasmic reticulum stress in diabetes mellitus. *Endocr. Rev.* **29**, 42–61 (2008).
- Mynatt, R. L. *et al.* Combined effects of insulin treatment and adipose tissue-specific agouti expression on the development of obesity. *Proc. Natl. Acad. Sci. U. S. A.* **94**, 919–922 (1997).

11. Kim, Y. Y. *et al.* Hepatic GSK3 β -Dependent CRY1 Degradation Contributes to Diabetic Hyperglycemia. *Diabetes* **71**, 1373–1387 (2022).
12. Lamia, K. A. *et al.* AMPK regulates the circadian clock by cryptochrome phosphorylation and degradation. *Science* (80-.). (2009) doi:10.1126/science.1172156.
13. Macotela, Y., Boucher, J., Tran, T. T. & Kahn, C. R. Sex and depot differences in adipocyte insulin sensitivity and glucose metabolism. *Diabetes* **58**, 803–812 (2009).
14. González-García, I., Gruber, T. & García-Cáceres, C. Insulin action on astrocytes: From energy homeostasis to behaviour. *J. Neuroendocrinol.* **33**, e12953 (2021).
15. Tahara, Y., Hirao, A., Moriya, T., Kudo, T. & Shibata, S. Effects of medial hypothalamic lesions on feeding-induced entrainment of locomotor activity and liver Per2 expression in Per2::luc mice. *J. Biol. Rhythms* **25**, 9–18 (2010).
16. Coppari, R. *et al.* The hypothalamic arcuate nucleus: a key site for mediating leptin's effects on glucose homeostasis and locomotor activity. *Cell Metab.* **1**, 63–72 (2005).
17. Mesaros, A. *et al.* Activation of Stat3 signaling in AgRP neurons promotes locomotor activity. *Cell Metab.* **7**, 236–248 (2008).
18. Wiater, M. F., Li, A. J., Dinh, T. T., Jansen, H. T. & Ritter, S. Leptin-sensitive neurons in the arcuate nucleus integrate activity and temperature circadian rhythms and anticipatory responses to food restriction. *Am. J. Physiol. Regul. Integr. Comp. Physiol.* **305**, (2013).
19. Huang, H. *et al.* ROCK1 in AgRP neurons regulates energy expenditure and locomotor activity in male mice. *Endocrinology* **154**, 3660–3670 (2013).
20. Wiater, M. F. *et al.* Circadian integration of sleep-wake and feeding requires NPY receptorexpressing neurons in the mediobasal hypothalamus. *Am. J. Physiol. - Regul. Integr. Comp. Physiol.* **301**, 1569–1583 (2011).
21. Méndez-Hernández, R., Escobar, C. & Buijs, R. M. Suprachiasmatic Nucleus-Arcuate Nucleus Axis: Interaction Between Time and Metabolism Essential for Health. *Obesity (Silver Spring)*. **28 Suppl 1**, S10–S17 (2020).
22. Cedernaes, J. *et al.* Transcriptional Basis for Rhythmic Control of Hunger and Metabolism within the AgRP Neuron. *Cell Metab.* (2019) doi:10.1016/j.cmet.2019.01.023.
23. Buijs, F. N. *et al.* Suprachiasmatic Nucleus Interaction with the Arcuate Nucleus; Essential for Organizing Physiological Rhythms. *eNeuro* **4**, (2017).
24. Kleinridders, A. *et al.* Insulin resistance in brain alters dopamine turnover and causes behavioral disorders. *Proc. Natl. Acad. Sci. U. S. A.* **112**, 3463–3468 (2015).
25. Smit, A. N., Patton, D. F., Michalik, M., Opiol, H. & Mistlberger, R. E. Dopaminergic regulation of circadian food anticipatory activity rhythms in the rat. *PLoS One* **8**, (2013).
26. Schoettner, K. *et al.* Characterization of Affective Behaviors and Motor Functions in Mice With a Striatum-Specific Deletion of Bmal1 and Per2. *Front. Physiol.* **13**, (2022).
27. Margolis, R. U. & Altszuler, N. Insulin in the Cerebrospinal Fluid. *Nat.* 1967 2155108 **215**, 1375–1376 (1967).
28. Iliff, J. J. *et al.* A paravascular pathway facilitates CSF flow through the brain parenchyma and the clearance of interstitial solutes, including amyloid β . *Sci. Transl. Med.* **4**, (2012).
29. Greco, A. V., Ghirlanda, G., Fedeli, G. & Gambassi, G. Insulin in the cerebro spinal fluid of man. *Eur. Neurol.* **3**, 303–307 (1970).
30. Scherer, T., Sakamoto, K. & Buettner, C. Brain insulin signalling in metabolic homeostasis and disease. *Nat. Rev. Endocrinol.* **17**, 468–483 (2021).
31. Rhea, E. M. *et al.* Blood-Brain Barriers in Obesity. *AAPS J.* **19**, 921 (2017).
32. Clegg, D. J., Riedy, C. A., Smith, K. A. B., Benoit, S. C. & Woods, S. C. Differential sensitivity to central leptin and insulin in male and female rats. *Diabetes* **52**, 682–687 (2003).
33. Nedungadi, T. P. & Clegg, D. J. Sexual dimorphism in body fat distribution and risk for cardiovascular diseases. *J. Cardiovasc. Transl. Res.* **2**, 321–327 (2009).
34. Shi, H., Sorrell, J. E., Clegg, D. J., Woods, S. C. & Seeley, R. J. The roles of leptin receptors on POMC neurons in the regulation of sex-specific energy homeostasis. *Physiol. Behav.* **100**, 165–172 (2010).
35. Mauvais-Jarvis, F., Clegg, D. J. & Hevener, A. L. The role of estrogens in control of energy balance and glucose homeostasis. *Endocr. Rev.* **34**, 309–338 (2013).
36. Miyata, S. Glial functions in the blood-brain communication at the circumventricular organs. *Front. Neurosci.* **16**, (2022).
37. Involvement of non-neuronal brain cells in AVP-mediated regulation of water space at the cellular, organ, and whole-body level - PubMed. <https://pubmed.ncbi.nlm.nih.gov/11070491/>.

38. Hablitz, L. M. *et al.* Circadian control of brain glymphatic and lymphatic fluid flow. *Nat. Commun.* **11**, (2020).
39. Barca-Mayo, O. *et al.* Astrocyte deletion of *Bmal1* alters daily locomotor activity and cognitive functions via GABA signalling. *Nature Communications* vol. 8 1–14 (Nature Publishing Group, 2017).
40. Wulff, K., Gatti, S., Wettstein, J. G. & Foster, R. G. Sleep and circadian rhythm disruption in psychiatric and neurodegenerative disease. *Nat. Rev. Neurosci.* *2010* **11**, 589–599 (2010).
41. Imamura, K. & Takumi, T. Mood phenotypes in rodent models with circadian disturbances. *Neurobiol. Sleep Circadian Rhythm.* **13**, (2022).
42. Qian, J. *et al.* Sex differences in the circadian misalignment effects on energy regulation. *Proc. Natl. Acad. Sci. U. S. A.* **116**, (2019).

REVIEWER COMMENTS

Reviewer #1 (Remarks to the Author):

The article entitled "Astrocytic insulin receptor controls circadian behavior via dopamine signaling in a sexually dimorphic fashion" submitted by Lopez and Barca-Mayo explores the role of astrocytic insulin receptor on light- and food-dependent entrainment of circadian rhythms, interestingly assessing a sex dimorphic effect.

In this revised version of the paper authors carefully addressed most of this reviewer's concerns about methodological aspects and data interpretation. There is still one major concern that has not been addressed and few minor changes suggested to be introduced before considering the article for publication.

Major concerns:

1) The genetic manipulation using the viral approach to knock-down IR in MBH (or VMH/ARC) astrocytes (now in Figure 7) is only partially validated. Besides successfully targeting the region authors should show that IR is actually down-regulated specifically in astrocytes. If this crucial validation experiment is not possible data interpretation need to be revised (i.e.: discussion section page 21 lines 607-632)

Minor concerns:

1) Page 2 line 52: has to read SCN

2) Page 3 line 61: Full names of BMAL1 and CLOCK are missing

3) Page 5 lines 128-132: I would avoid any reference to the in vivo situation and delete this new paragraph. As mentioned during the first round of revision, the in vitro data from Fig 1 is not sufficient to make any kind of conclusion of such a complex interplay, to begin with, DEX is a synthetic glucocorticoid with a different receptor binding capacity and stability than endogenous hormones. Moreover, a combined DEX+INS treatment is missing.

4) Page 10 line 293: "that" is repeated

Reviewer #2 (Remarks to the Author):

Authors did a thorough assessment and replied to all my concerns. Hence, I have no further

inquiries.

Reviewer #3 (Remarks to the Author):

The authors have addressed all concerns.

REVIEWER#1

Overall comment: The article entitled "Astrocytic insulin receptor controls circadian behavior via dopamine signaling in a sexually dimorphic fashion" submitted by Lopez and Barca-Mayo explores the role of astrocytic insulin receptor on light- and food-dependent entrainment of circadian rhythms, interestingly assessing a sex dimorphic effect.

In this revised version of the paper authors carefully addressed most of this reviewer's concerns about methodological aspects and data interpretation. There is still one major concern that has not been addressed and few minor changes suggested to be introduced before considering the article for publication.

Response: We would like to extend our sincere gratitude for your careful review of our article. Your feedback and constructive suggestions have played a crucial role in enhancing the quality and clarity of our manuscript. We have diligently considered your comments and suggestions, and we are pleased to provide a detailed point-by-point response to address your concerns.

MAJOR CONCERNS

Reviewer's major concern 1: The genetic manipulation using the viral approach to knock-down IR in MBH (or VMH/ARC) astrocytes (now in Figure 7) is only partially validated. Besides successfully targeting the region authors should show that IR is actually down-regulated specifically in astrocytes. If this crucial validation experiment is not possible data interpretation need to be revised (i.e.: discussion section page 21 lines 607-632).

Response: We sincerely appreciate your thorough evaluation of our manuscript and your concern regarding the validation of our viral approach for knocking down IR in MBH (or VMH/ARC) astrocytes, as presented in Figure 7.

Recognizing the critical importance of demonstrating specific down-regulation of IR in astrocytes, we encountered technical challenges with the antibodies, which, unfortunately, prevented us from directly confirming the knockout of IR β in MBH/VMH/ARC astrocytes by immunofluorescence. However, we have now conducted a Western blot analysis to confirm the knockout of IR β in the MBH and ARC of the mutants (see **New Figures 7A and 7F**). This new experimental data, along with GFP labeling demonstrating specific infection within the MBH and ARC, provides additional robust evidence for the downregulation of IR in the targeted astrocytes. Moreover, to strengthen the evidence for IR β knockout, we have referenced our prior publication by García-Cáceres et al. (Cell, 2016¹), wherein we employed the same viral approach and methodology to achieve knockout efficiency (**Page 17, lines 489-495 of the new version of the manuscript**). In that study, we demonstrated a significant reduction in IR β mRNA expression in the MBH of the mutants. Additionally, we observed a decrease in the percentage of phosphorylated AKT-positive astrocytes in the MBH after peripheral INS injection. This reference provides historical context and further substantiates the effectiveness of our approach in knocking down IR β . Furthermore, we have included this reference and discussed its relevance in the discussion section (**Page 21, lines 613-619 of the new version of the manuscript**) and revised the data interpretation.

We genuinely appreciate your understanding of the complexities inherent in scientific research and the patience you've shown in guiding us through the review process. Your insightful feedback has played a pivotal role in enhancing the rigor and quality of our manuscript.

MINOR CONCERNS

Reviewer's minor concern 1: Page 2 line 52: has to read SCN.

Response: Thank you for pointing out the typographical error on Page 2, line 52. The correction has been made to read "SCN."

Reviewer's minor concern 2: Page 3 line 61: Full names of BMAL1 and CLOCK are missing.

Response: We appreciate your observation. The full names of BMAL1 and CLOCK have been included for clarity on Page 3, line 61.

Reviewer's minor concern 3: Page 5 lines 128-132: I would avoid any reference to the in vivo situation and delete this new paragraph. As mentioned during the first round of revision, the in vitro data from Fig 1 is not sufficient to make any kind of conclusion of such a complex interplay, to begin with, DEX is a synthetic glucocorticoid with

a different receptor binding capacity and stability than endogenous hormones. Moreover, a combined DEX+INS treatment is missing.

Response: We acknowledge your valid concerns and agree with your suggestion to remove the paragraph on Page 5, lines 128-132, which discussed the in vivo situation and interplay between factors. We understand the limitations of in vitro data and the need for a combined DEX+INS treatment to draw more meaningful conclusions.

Reviewer's minor concern 4: Page 10 line 293: "that" is repeated.

Response: Thank you for catching the repetition. We have removed the redundant "that" on Page 10, line 290 of the new version of the manuscript, to improve the sentence's clarity and readability.

We appreciate your meticulous review of our manuscript and your valuable feedback. Your input has contributed to the refinement of our work, and we hope these revisions address your concerns adequately.

REFERENCES

1. García-Cáceres, C. *et al.* Astrocytic Insulin Signaling Couples Brain Glucose Uptake with Nutrient Availability. *Cell* **166**, 867–880 (2016).

REVIEWERS' COMMENTS

Reviewer #1 (Remarks to the Author):

In this revised version of the paper authors carefully addressed all reviewer's concerns about methodological aspects and data interpretation.

REVIEWER#1

Overall comment: In this revised version of the paper authors carefully addressed all reviewer's concerns about methodological aspects and data interpretation.

Response: We would like to express our sincere appreciation to Reviewer #1 for dedicating valuable time to review our manuscript and for providing insightful feedback. We are delighted that our efforts in addressing their excellent concerns have led to a significant improvement in the manuscript.